# MixtureVitae: Open Web-Scale Pretraining Dataset With High Quality Instruction and Reasoning Data Built from Permissive-First Text Sources

Huu Nguyen[1,3,4,*], Victor May[1,*], Harsh Raj[1,2,3,4,5,17,*], Marianna Nezhurina[1,3,4,5,17], Yishan Wang[1,6], Yanqi Luo[7], Minh Chien Vu[8], Taishi Nakamura[4,9], Ken Tsui[4,15], Van Khue Nguyen[4,10], David Salinas[11,12,17], Aleksandra Krasnodębska[13], Christoph Schuhmann[3], Mats Leon Richter[14], Xuan-Son Vu[16], and Jenia Jitsev[1,3,4,5,17]

[*]Equal contribution
 *Correspondence to: huu@ontocord.ai*
[1]Ontocord
[2]Northeastern University
[3]LAION
[4]Open-Ψ (Open-Sci) Collective
[5]Juelich Supercomputing Center (JSC), Research Center Juelich (FZJ)
[6]Carnegie Mellon University
[7]Salesforce
[8]Detomo Inc.
[9]Institute of Science Tokyo
[10]École Polytechnique, IP Paris
[11]ELLIS Institute Tuebingen
[12]University of Freiburg
[13]NASK
[14]Montreal Institute for Learning Algorithms, University of Montreal, Université de Montréal
[15]Independent Researcher
[16]Lund University and DeepTensor AB
[17]openEuroLLM Team

**Reviewed on OpenReview:** https://openreview.net/forum?id=SyCcUNUUMf

## Abstract

We present **MixtureVitae**, an open-access pretraining corpus[1] built to minimize legal risk while providing strong downstream performance. **MixtureVitae** follows a permissive-first, risk-mitigated sourcing strategy that combines public-domain and permissively licensed text (e.g., CC-BY/Apache) with carefully justified low-risk additions (e.g., government works and EU TDM-eligible sources). **MixtureVitae** adopts a simple, single-stage pretraining recipe that integrates a large proportion of permissive synthetic instruction and reasoning data—signals typically introduced during post-training and generally scarce in permissive web corpora. We categorize all sources into a three-tier scheme that reflects varying risk levels and provide shard-level provenance metadata to enable risk-aware usage. In controlled experiments using the open-sci-ref training protocol (fixed architectures and hyperparameters; 50B and 300B token budgets across 130M–1.7B parameters), models trained on **MixtureVitae** consistently outperform other permissive datasets across a suite of standard benchmarks, and at the 1.7B-parameters/300B-tokens setting, they match FineWeb-Edu and approach DCLM–demonstrating that the large fraction of reasoning and instruction data does not come at the cost of general-purpose language understanding. Performance is particularly strong on

---

[1]Dataset, source code for experiments reproduction and pre-trained models are available at **MixtureVitae** repo.

MMLU and on math and code benchmarks: a 1.7B model pretrained on 300B MixtureVitae tokens outperforms all strong non-permissive reference datasets and matches or exceeds smolLM2-Instruct, a strong 1.7B instruction-tuned baseline on GSM8K, HumanEval, and MBPP, despite using over $36\times$ fewer tokens (300B vs. $\approx$11T). Supported by a thorough decontamination analysis, these results show that permissive-first data with high instruction and reasoning density, tiered by licensing and provenance-related risk, can provide a practical and risk-mitigated foundation for training capable LLMs, reducing reliance on broad web scrapes without sacrificing competitiveness.

# 1 Introduction

The proliferation of large language models (LLMs) has transformed the landscape of artificial intelligence, yet their development often relies on a legally and ethically precarious foundation. The vast majority of performant models are pretrained on massive web scrapes, indiscriminately mixing public-domain content with copyrighted materials such as books, news articles, and personal websites without explicit permission (Raffel et al., 2020; Gao et al., 2020). This practice has led to a growing number of copyright infringement lawsuits, creating significant legal uncertainty for both academic researchers and commercial developers and threatening the future of the field. At the same time, practitioners who wish to avoid this risk have few alternatives, as most high-performing pretraining mixtures rely, at least in part, on opaque or non-permissive web scrapes.

Compounding this uncertainty is the prevailing assumption that state-of-the-art performance is inextricably linked to the sheer scale and diversity offered by these legally ambiguous web scrapes. The absence of a high-performance, large-scale pretraining dataset that actively mitigates these risks has forced a difficult choice between performance and compliance. In practice, the strongest open baselines such as FineWeb-Edu (Penedo et al., 2024) and DCLM (Li et al., 2024) still rely on mixed-license or unspecified web data, whereas strictly permissive corpora tend to lag behind them on reasoning-heavy benchmarks. This raises a critical question: **Can a powerful general-purpose language model be trained on a dataset that provides a more legally robust foundation, relying heavily on permissive sources?**

To this question, we answer "yes": We introduce MixtureVitae, a **422**-billion-token, open-access pretraining dataset constructed to minimize copyright risk while explicitly demonstrating that a reasoning- and instruction-dense, permissive-first mixture can substantially close the performance gap to leading non-permissive corpora. The core of MixtureVitae's "permissive-first" data comprise (1) text with clear and permissive licenses (e.g., CC-BY-*, Apache 2.0), public-domain text, and copyright-exempt text such as US federal works (see Appendix H) and (2) risk-mitigated text. Following Phi-4 (Abdin et al., 2024), which shows that the addition of synthetic and web-rewrite data boosts performance, we address the scarcity of real, human-written reasoning and conversational dialogue in strictly permissive sources by significantly augmenting MixtureVitae with targeted synthetic data, which is derived from permissive models and sources. We call this combination of expressly licensed and risk-mitigated methods the **"permissive-first"** approach.

To validate our approach, we train models with **130M, 400M, 1.3B, and 1.7B parameters** on MixtureVitae and compare their performance against several prominent open datasets. The results first confirm that MixtureVitae **significantly outperforms all other permissively licensed baselines**, with the performance gap widening as the model scale increases. Crucially, MixtureVitae's large fraction of reasoning and instruction data does not sacrifice general-purpose language understanding–it outperforms permissive baselines and matches leading non-permissive corpora on standard benchmarks while *simultaneously* achieving dramatically stronger reasoning, math, and code performance. The more critical test, however, is against strong non-permissive datasets containing higher proportions of copyrighted or ambiguously-licensed material. In this setting, our models achieve competitive performance, and on math and code benchmarks, our 1.7B base model outperforms all strong non-permissive baselines and matches or exceeds a strong 1.7B instruction-tuned baseline (SmolLM2 (Ben allal et al., 2025)) despite being trained on a dramatically smaller budget (over $36\times$ fewer tokens).

In summary, our contributions are threefold:

**Permissive-first, risk-mitigated, and performant recipe for pretraining corpora.** We present **MixtureVitae**, the first highly-performant, permissive-first, and risk-mitigated pretraining corpus that deliberately front-loads high-quality reasoning and instruction data to drive capability gains in small models. It is organized into auditable provenance tiers and constructed via a positive-inclusion pipeline, avoiding the need for retroactive filtering.

**We demonstrate that reliance on indiscriminately scraped, high-risk copyrighted data is not a prerequisite for training capable LLMs.** Leveraging the `open-sci-ref` (Nezhurina et al., 2025) protocol to ensure rigorous comparison across 130M–1.7B parameter scales, we demonstrate the value of front-loading instruction and reasoning data into pre-training. Our **422**B-token, permissive-first mixture closes the gap to mixed-license baselines while providing an auditable legal provenance. Furthermore, we show that our 1.7B base model, despite a limited 300B token budget, is comparable across multiple reasoning benchmarks to a strong 1.7B instruction-tuned baseline—trained on roughly 36× more tokens (≈11T).

**Evaluation integrity and reusable artifacts**. We perform a large-scale 13-gram decontamination analysis across all benchmarks, showing that **MixtureVitae**'s gains persist on decontaminated test sets and when removing shards responsible for most detected overlap, and we release the corpus, shard-level provenance metadata, and curation code to enable compliant, reproducible pretraining in future work.

## 2  Dataset

We adopt a permissive-first, risk-mitigated strategy, combining sources with clear permissive licenses (e.g. CC-BY, Apache, public domain) with narrowly justified inclusions (government works, EU TDM-eligible data) and targeted synthetic data. Within this framework, the **MixtureVitae** dataset is constructed from three primary categories: curated sources for domain-specific expertise, diverse web data for language and general knowledge and instruction-following and reasoning datasets to enhance reasoning and task-completion abilities.

The major categories of our corpus are visualized in Figure 1a. We provide a granular breakdown showing the token count for each component (Figure 6), the license distribution (Figure 1b), and synthetic data usage (Figure 2a). Specific data sources are detailed in the following subsections.

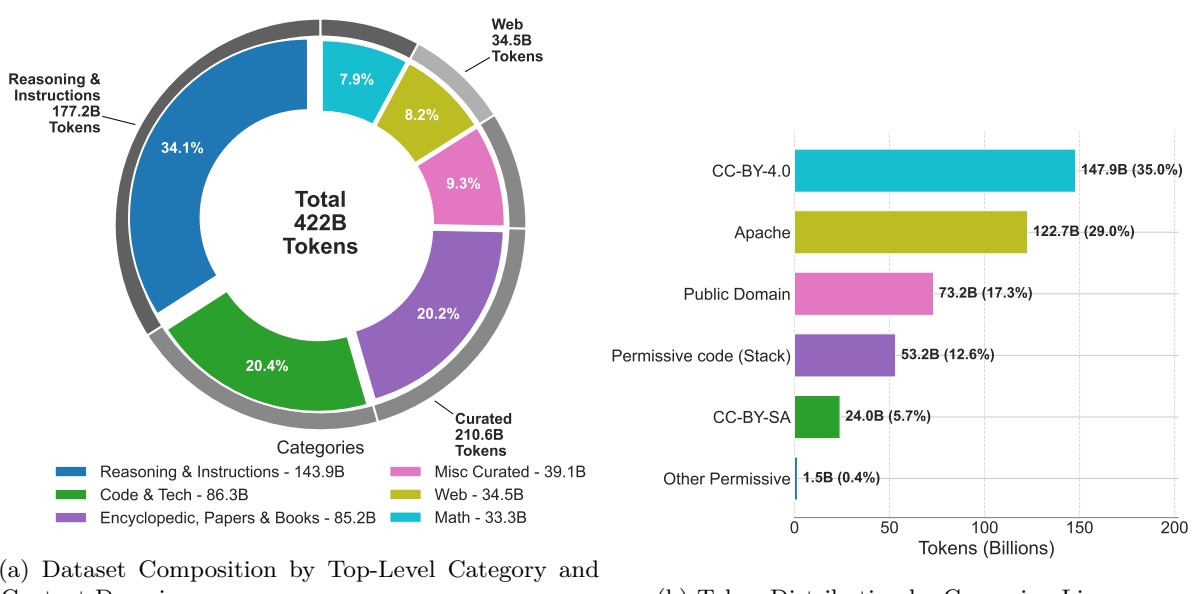

(a) Dataset Composition by Top-Level Category and Content Domain

(b) Token Distribution by Governing License

Figure 1: Composition of the **MixtureVitae** dataset (permissive-first, risk-mitigated composition).

## 2.1 Data Sources

Our dataset selection process is governed by a two-layer criteria, prioritizing risk mitigation followed by quality and capability objectives:

**Legal & Licensing:** The primary filter is legal compliance. A dataset is considered only if it operates under a clear permissive license (e.g., CC-BY, Apache 2.0) or is in the public domain. For synthetic data, we further scrutinize the provenance of seed corpora and generator models (Appendix K). The majority of our synthetic sources satisfy full provenance transparency (classified as Tier 1), while a minority of community reasoning datasets with opaque provenance are categorized as Tier 2 to manage residual risk.

**Quality & Capability:** Among compliant sources, we prioritize datasets with prior evidence of high performance in community mixtures (e.g., Soldaini & Lo, 2023). Furthermore, to address the reasoning deficits typical of strictly permissive web scrapes, we target high-density instruction and reasoning data, a choice driven by the need to boost performance on tasks such as GSM8K (Cobbe et al., 2021) and MMLU(Hendrycks et al., 2021a).

The following sections describe each of the three categories of data in MixtureVitae: web, curated sources, and instruction and reasoning datasets.

### 2.1.1 Web-Scale Corpora

One subset of our pre-training data is derived from web-scale datasets including Nemotron-CC (Su et al., 2025), MGACorpus (Hao et al., 2025), and FineFineWeb (M-A-P et al., 2024). It also contains synthetic data generated by rephrasing web text from Nemotron-CC and MGACorpus.

### 2.1.2 Curated Datasets

To incorporate domain-specific knowledge and high-quality text, we curate diverse sources: public financial documents from SEC EDGAR (U.S. Securities and Exchange Commission, 2024), multilingual encyclopedic articles from MegaWika (Barham et al., 2023) and TxT360 (Tang et al., 2024), scientific papers from arXiv (Clement et al., 2019) and peS2o (Soldaini & Lo, 2023), medical data from Pubmed (National Library of Medicine (U.S.), 1996), code from The Stack v1 (Kocetkov et al., 2023), patents from the USPTO database (United States Patent and Trademark Office, 2024) and EuroPat (Heafield et al., 2022), mathematical problems from Deepmind Math (Saxton et al., 2019), and video transcripts from both VALID (Nguyen et al., 2024) and the YouTube Commons corpus (Langlais, 2024), news and law data from the Open License Corpus (Min et al., 2024). We source 12.6% of our dataset from **The Stack v1**, a permissive-first, risk-mitigated code dataset governed by the OpenRAIL-M license. We discuss its permissiveness situation in Appendix I.

### 2.1.3 Instruction and Reasoning Datasets

To enhance instruction-following and reasoning, we follow Abdin et al. (2024) by including considerable synthetic and web-rewrite data. We extensively use fully and partially synthetic data – all generated from permissive or public-domain seed data using models under permissive licenses.

**General Instruction Following** We include a strong instruction-following baseline with the Magpie Collection (Xu et al., 2024), its derivatives (e.g., Magpie-Phi3-Pro). This is augmented with preference data from UltraFeedback (Cui et al., 2024) and NVIDIA's SFT data blend NVIDIA (2024), which contains a curated mixture of permissively licensed subsets from public datasets, including OASST (Köpf et al., 2023), CodeContests (Li et al., 2022), FLAN (Chung et al., 2022), OpenPlatypus (Lee et al., 2023), and the training split of GSM8K (Cobbe et al., 2021). Additionally, we augment the P3 (Sanh et al., 2022) dataset with a few-shot and multiple-choice format.

**Reasoning** To improve reasoning, we incorporate general corpora such as Glaive-AI Reasoning Dataset (Glaive AI, 2023) and OpenThoughts (Guha et al., 2025) as well as domain-specific datasets: the legal dataset CaseHOLD (Zheng et al., 2021), scientific Q&A from the OpenScience collection (NVIDIA Corporation, 2025), and agent-focused instructions from OpenManus-RL (Ulab-UIUC and MetaGPT, 2024).

**Mathematics and Coding** To strengthen quantitative reasoning, we combine our internally developed synthetic Math Word Problems dataset (Appendix G) with established datasets like MetaMathQA (Yu et al., 2024) and DM-Math (Saxton et al., 2019), further enriched with large-scale math instruction sets, including OpenMathInstruct-2 (Toshniwal et al., 2024b), DART-MATH (Tong et al., 2024), Nemo-Math (Mahabadi et al., 2025), and Prism-Math (NVIDIA, 2025). For coding, we combine the Ling Coder collection Codefuse Team et al. (2025) with executable instructions from the StarCoder dataset Kocetkov et al. (2023) to target a wide range of software engineering tasks.

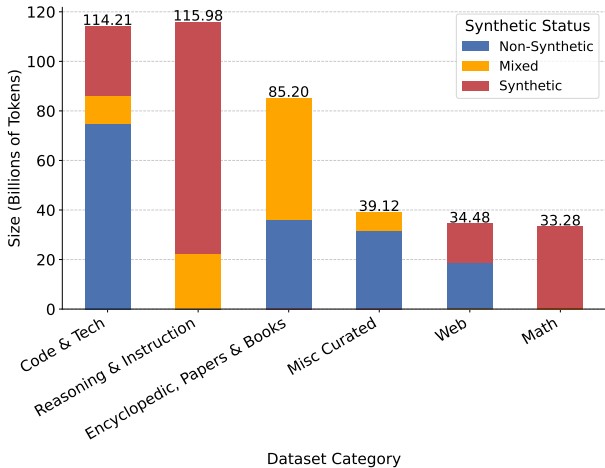

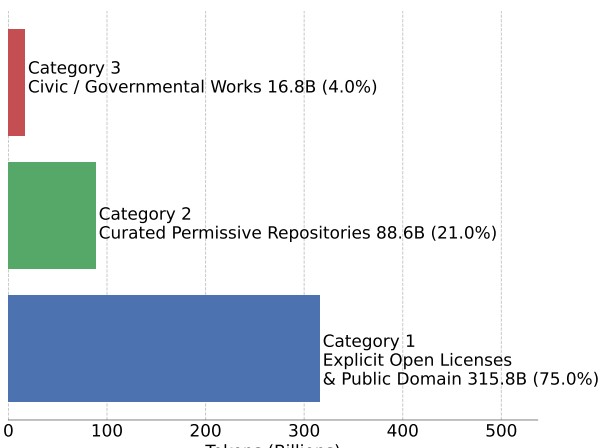

(a) **MixtureVitae** composition by origin (total token counts at the top in billions). Each bar represents one of the six primary content domains (as in Figure 1a), segmented by source type: **Non-Synthetic** (real human-written text and code), **Mixed** (sources with partial synthetic data), and **Synthetic** (data generated by permissive models from permissive seeds).

(b) Legal provenance and risk-mitigation tiers of the **MixtureVitae** corpus. The dataset is segmented into its three constituent legal categories, with all sources falling into a permissive-first or risk-mitigated tier. Token counts (billions) and total corpus percentages are shown for each category.

Figure 2: Composition and provenance of **MixtureVitae**: **(a)** Synthetic-status distribution across the six content domains, **(b)** licensing tiers and risk posture for the corpus.

### 2.1.4 Licensing Tiers and Risk Profiles

To make the provenance and legal footing of **MixtureVitae** transparent, we conceptualize all dataset components into *tiers* based on license type and expected risk profile (see Figure 2b and Table 15).[2]

**Tier 1 − Explicit Open Licenses & Public Domain.** This tier encompasses text and code under clear permissive licenses (e.g., CC0, CC-BY, Apache 2.0, MIT, BSD, a permissive subset of P3) or in the public domain, such as encyclopedic resources, scientific papers, and portions of curated math corpora. Because licenses are explicit and permissive, the legal risk of reuse is minimal. This tier also includes synthetic data generated from permissively licensed models and seed data.

**Tier 2 − Curated Permissive Corpora with Upstream Opacity.**

**(a) Permissive Corpora With Partial or Unverified Provenance.** This subset includes resources such as THE STACK V1 and Wikipedia-derived corpora. The released dataset all carries a permissive license, and curators apply filters (e.g., repository-level license heuristics). However, because provenance is only partially tracked at the file or example level, there remains some residual uncertainty about the licensing status of individual items, hence its separation from Tier 1. This Tier also includes datasets that have no license, but

---

[2]The high-level groupings presented in this section (e.g., "Code & Tech", "Reasoning") and the shard breakdowns in the Appendices are primarily organizational abstractions for visualization and provenance tracking. In practice, the actual training data construction follows a granular *domain-aware mixing strategy* (detailed in Section 2.2.4), where documents are clustered by base URL or provenance to preserve domain coherence per sample, rather than strictly sampling from rigid high-level partitions.

the underlying data is public domain or permissive and requiring the same license as the upstream data, or where the data is solely obtained synthetically from a model that is permissively licensed.

**(b) Synthetic Data with Non-Permissive or Unverifiable Generators or Seeds.** This tier contains datasets that are themselves permissively licensed (e.g., Apache/MIT/CC-BY), but where either (i) the generator model used to create the synthetic data operates under a more restrictive license (e.g., Llama-3 community license, OpenAI API terms), or (ii) the seed data contains slices whose provenance cannot be fully audited (e.g., partially opaque community mixtures). These datasets constitute only $\approx4\%$ of **MixtureVitae** and are isolated for transparency so that users who require a strictly permissive generator and seed provenance can exclude them (more detail in Table 15).

**Tier 3 – Civic / Governmental Works.** This tier includes materials that are either statutory public domain (e.g., U.S. federal works) or under a strong public-purpose rationale for reuse (e.g., government websites, regulatory notices). While not always explicitly licensed, such work–typically created for dissemination–is widely recognized as low-risk for inclusion. Filtering with copyright keyword checks further reduces the possibility of inadvertently including restricted content.

We emphasize that this tiering is **risk-mitigated rather than risk-free** and rests primarily on U.S. fair-use doctrine and the EU TDM exception; copyright regimes differ across jurisdictions, so international adopters should consult local counsel before deploying Tier 2/Tier 3 subsets in settings where their permissibility is jurisdiction-specific (Appendix H).

## 2.2 Data Processing Pipeline

To transform the raw data sources into a high-quality and permissively licensed pretraining corpus, we develop a multistage data processing pipeline. Our curation pipeline includes the following stages: ensuring permissive licensing, filtering for CSAM and offensive language, improving overall content quality, and reducing data redundancy. The following sections detail each component.

### 2.2.1 Permissiveness Filtering

In contrast to standard data pipelines that rely on the retroactive negative filtering of broad web scrapes (e.g., Fan et al., 2025), we employ a **positive inclusion** strategy for web data. Rather than ingesting broad web dumps and filtering post-hoc, we positively select sources based on auditable permissive status. Specifically, we (i) apply an explicit allowlist of governmental and international domains (Appendix J.1), (ii) curate a set of websites with known permissive licenses (Appendix J.2), and (iii) expand this set with risk-mitigated documents by searching for permissive license keywords (e.g., "CC-BY-SA"), excluding documents with restrictive terms (e.g., "all rights reserved"). This upfront design minimizes the risk of including paywalled or opted-out content (e.g., commercial news). We justify the inclusion of governmental works under a strong fair-use rationale, considering their public purpose, content type, and minimal market impact (Appendix H).

### 2.2.2 Quality and Safety Filtering

Per standard practices (Raffel et al., 2020), we remove documents with base64-encoded text (which can disrupt training) and duplicative headers and footers (e.g., "Home | Search") from FineFineWeb. We remove obscene, adult and CSAM-related content with keyword-based blocklists adapted from prior work (Laurençon et al., 2022; Nakamura et al., 2025). For Wikipedia-based documents, we remove articles about films, sporting events, and biographies of living persons in English with applied targeted filtering, to minimize memorization of facts about people, in case of objection to incorrect facts about people being generated by models trained on **MixtureVitae**. Besides dataset-level filters, we also evaluate the final model's safety profile via standard red-teaming (Appendix E.5).

### 2.2.3 Deduplication

Informed by recent findings in large-scale data curation, our deduplication strategy prioritizes diversity over purity. While removing exact repetitions mitigates harmful memorization (Lee et al., 2022), prior research finds that aggressive, global near-duplicate removal can be detrimental. For example, the creators

of the **FineWeb-Edu** dataset (Penedo et al., 2024) reported *worsened* model performance by global fuzzy deduplication, postulating that it removed "too much quality data."

Therefore, we adopt a local-only approach. We first apply **intra-dataset deduplication** using prefix-based exact matching to remove verbatim boilerplate text (Lee et al., 2022). We **intentionally avoid full, cross-dataset fuzzy deduplication** to preserve near-duplicates (e.g., Wikipedia articles with different formatting across sources). We posit that doing so retains **"stylistic and domain diversity,"** a factor shown to be helpful for model generalization (Chen et al., 2024).

### 2.2.4 Training Example Curation

Our process for creating training examples involves several stages:

**1. Heuristic Cleaning:** We remove boilerplate content by eliminating repetitive n-gram prefixes and suffixes, following standard web data cleaning pipelines (Raffel et al., 2020).

**2. Fine-grained Deduplication:** To enhance data quality, we segment documents into sentences and remove duplicate sentences within each document. Documents with high internal repetition (sentence duplication rate $> 75\%$) are discarded entirely, as this has been shown to improve model performance (Lee et al., 2022).

**3. Domain-Aware Mixing:** To construct the final training examples, we employ a domain-aware data mixing strategy (Xie et al., 2023). Documents are clustered by their base URL (a proxy for domain), and sentences are concatenated first within their original document, then packed with other documents from the same cluster.

### 2.2.5 Additional Filtering for Synthetic Datasets

To ensure that the synthetic subsets of **MixtureVitae** adhere to our permissive-first, risk-mitigated approach, we prioritize data originating from seeds that are sourced from permissive sources and generated with models that are themselves permissively licensed. A small portion ($\approx 4\%$) of **MixtureVitae** originates from sources with restricted, mixed, or opaque provenance and is isolated into Tier 2(b), as detailed in Appendix K and Table 15.

## 3 Experiments

We empirically validate the efficacy of **MixtureVitae** through a comprehensive set of evaluations. We begin by outlining our controlled experimental framework, model architectures, and baseline selection in Section 3.1. We then present the primary scaling behavior and general benchmark performance in Section 3.2, followed by a focused evaluation on reasoning, mathematics, and coding tasks in Section 3.3. To ensure the integrity of these findings, we detail our decontamination protocol and leakage analysis in Section 3.4. Finally, we isolate the contributions of specific dataset components through ablation studies in Section 3.5, highlighting the critical impact of instruction and reasoning data density.

### 3.1 Experimental Setup

To empirically validate the quality of the **MixtureVitae** pretraining dataset, we conduct a large-scale comparative study against a selection of prominent open pretraining datasets. We isolate the impact of the dataset on downstream performance using the **open-sci-ref** training procedure (Nezhurina et al., 2025), which enables systematic control of factors affecting benchmark scores. As in `open-sci-ref`, we fix the model architecture (Table 4, sizes: 0.13B, 0.4B, 1.3B, 1.7B) and training hyperparameters (Table 5), varying only the dataset. This design ensures that any performance difference can be attributed solely to the dataset.

Also, following the numbers given in `open-sci-ref`, we train each model on two token budgets: 50B and 300B, to analyze scaling effects. Conducting separate training runs on each budget, rather than using intermediate checkpoints, thus ensuring a consistent data distribution and allowing for proper optimization of learning rate schedules for each specific token budget (Hoffmann et al., 2022). This follows standard practice: Data mixtures effective at small token budgets may not generalize to larger ones (Albalak et al., 2023).

To guard against test-set leakage, we also perform a large-scale 13-gram decontamination analysis and re-evaluation; Section 3.4 and Appendix F detail this procedure.

Within this controlled evaluation framework, we compare **MixtureVitae** with the set of public baselines evaluated in `open-sci-ref`, with the addition of a representative selection of permissively licensed datasets. As detailed in Table 3, the comparison set includes two groups:

- **Non-Permissive/Mixed-License Baselines.** C4 (Raffel et al., 2020), The Pile (Gao et al., 2020), SlimPajama (Shen et al., 2024), FineWeb-Edu (Penedo et al., 2024), Nemotron-CC-HQ (Su et al., 2025), DCLM-baseline (Li et al., 2024), HPLT Monolingual Datasets v2.0 (Burchell et al., 2025);

- **Permissive Baselines.** the English subset of CommonCorpus (Langlais et al., 2025), as well as Comma-0.1, training dataset derived from CommonPile (Kandpal et al., 2025).

All datasets are tokenized using the GPT-NeoX-20B tokenizer (Black et al., 2022), resulting in a vocabulary size of 50,304. The models are trained using Megatron-LM (Shoeybi et al., 2020), and the evaluations are performed using LM Evaluation Harness (Gao et al., 2021).

Model performance is evaluated on recognized downstream task benchmarks: MMLU (Hendrycks et al., 2021a), COPA (Roemmele et al., 2011), LAMBADA (Paperno et al., 2016), OpenBookQA (Mihaylov et al., 2018), Winogrande (Sakaguchi et al., 2021), ARC (Challenge and Easy) (Clark et al., 2018), BoolQ (Clark et al., 2019), HellaSwag (Zellers et al., 2019), Commonsense-QA (Talmor et al., 2019) and PIQA (Bisk et al., 2020).

## 3.2 Experiment Results

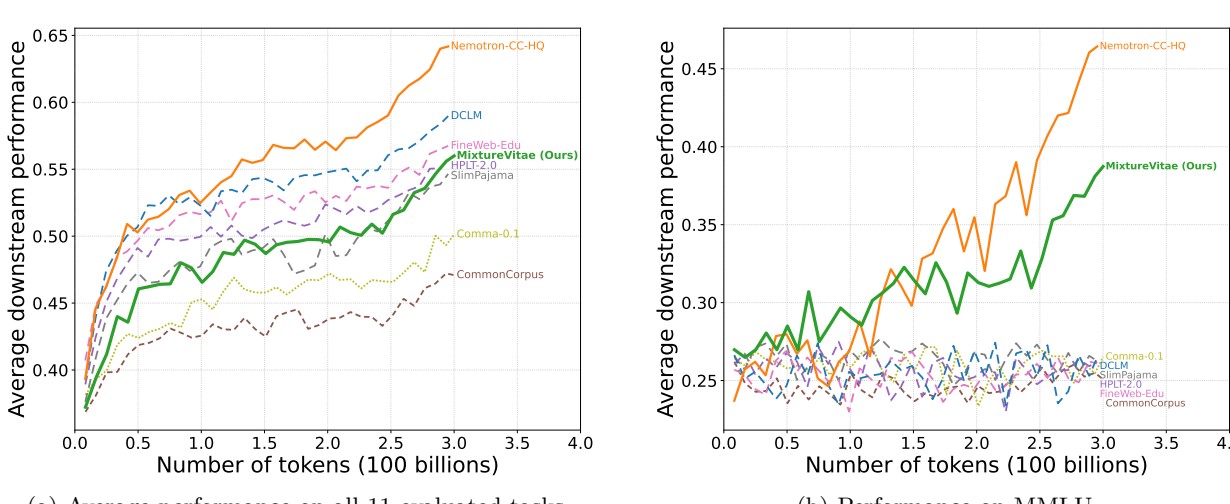

(a) Average performance on all 11 evaluated tasks.

(b) Performance on MMLU.

Figure 3: Performance comparison of pretraining datasets for a 1.7B-parameter model trained up to a 300B token budget, showing downstream accuracy as a function of the number of training tokens.

**Overall average performance.** At a 300B-token budget, **MixtureVitae** shows strong performance when compared to the reference permissive datasets and is almost comparable to the non-permissive datasets (Figure 3, Tab. 1). **MixtureVitae** outperforms all permissive dataset baselines by a significant margin, with gaps widening considerably for larger model sizes, in terms of average performance across all 10 tasks (see Figure 3a, Tab. 1). Non-permissive datasets, particularly Nemotron-CC-HQ and DCLM, still achieve the highest overall performance. Approaching the 300B token budget, **MixtureVitae** catches up to FineWeb-Edu and DCLM. More importantly, while the top-performing models are still trained on non-permissive datasets like Nemotron-CC-HQ and DCLM, our results demonstrate that this performance gap is no longer an inevitability. **MixtureVitae** proves that a dataset built on a fully permissive, risk-mitigated foundation can

Table 1: Performance comparison of 1.7B-parameter models trained on different pretraining datasets with a 300B token budget. *Italic* denotes the best result among permissive-only datasets, while **bold** indicates the best result overall, including mixed-license datasets. MixtureVitae outperforms other permissive datasets across most benchmarks. On reasoning related MMLU, BoolQ, and CommonSense-QA, it also outperforms strong non-permissive baselines.

| Benchmark | MixtureVitae (permissive) | Comma-0.1 (permissive) | CommonCorpus (permissive) | FineWeb-Edu (mixed-license) | DCLM (mixed-license) |
|---|---|---|---|---|---|
| COPA | *0.72* | 0.71 | 0.71 | 0.82 | **0.89** |
| Lambada | 0.49 | *0.54* | 0.49 | 0.56 | **0.69** |
| OpenBookQA | *0.36* | 0.33 | 0.31 | **0.42** | 0.39 |
| Winogrande | 0.59 | *0.60* | 0.56 | 0.62 | **0.65** |
| MMLU | ***0.39*** | 0.27 | 0.25 | 0.26 | 0.26 |
| ARC-Challenge | *0.41* | 0.36 | 0.32 | **0.48** | 0.44 |
| ARC-Easy | *0.72* | 0.63 | 0.61 | **0.77** | 0.75 |
| BoolQ | ***0.73*** | 0.62 | 0.62 | 0.65 | 0.72 |
| CommonSense-QA | ***0.49*** | 0.21 | 0.19 | 0.21 | 0.20 |
| HellaSwag | *0.55* | 0.53 | 0.45 | 0.67 | **0.71** |
| PIQA | *0.71* | *0.71* | 0.66 | 0.77 | **0.78** |
| Average | *0.56* | 0.50 | 0.47 | 0.57 | **0.59** |

achieve highly competitive results—significantly outperforming all other permissive baselines and landing within a small, practical margin of top-tier, legally-ambiguous corpora. This finding directly challenges the prevailing assumption that reliance on high-risk, indiscriminately scraped copyrighted data is a prerequisite for training capable LLMs. MixtureVitae performs particularly well relative to others on reasoning related tasks like **MMLU** (Figure 3b, Tab. 1), where most baselines are near random chance. Among all the baselines, only Nemotron-CC-HQ catches up to MixtureVitae at around 260B and overtakes it past that point. Our findings also hold at the 50B token budget scale (App. Sec. E.2). For scaling behavior on language tasks with compute, see App. Sec. E.3.

**Performance on single tasks.** We show performance on each single task in Tab. 1 and in the App. Sec. E.1 (App. Fig. 7). MixtureVitae outperforms other permissive datasets on MMLU, Arc Challenge, Arc Easy and BoolQ, while closely matching DCLM and FineWeb-Edu. On PIQA, HellaSwag, Winogrande, OpenBookQA, MixtureVitae is on par with Comma-0.1, while both are behind non-permissive datasets. Lambada is the only task where MixtureVitae falls behind Comma-0.1. We thus observe MixtureVitae to be particularly strong on reasoning-related tasks.

## 3.3 Results on Problem Solving and Instruction-Based Downstream Tasks

To further demonstrate the performance of the MixtureVitae dataset, we evaluate the model on a set of math, code, and instruction benchmarks: GSM8k (Cobbe et al., 2021), MBPP (Austin et al., 2021), IF-Eval (Zhou et al., 2023). Our evaluation uses the final 1.7B model checkpoints after training for 300B tokens using the `open-sci-ref` protocol (exact evaluation setup in App. Tab. 7).

Unlike traditional web-only baselines (e.g., C4, FineWeb, DCLM), MixtureVitae utilizes a *reasoning and instruction-heavy* pretraining mixture. Compared against base models with same architecture and matched training compute, this front-loading strategy shows capabilities typically associated with post-training. This pretraining composition leads to a more token-efficient and simple path to reasoning competence already after single base model pre-training stage, matching or outperforming conventional multi-stage extensive pre- and post-training procedures.

The results (Table 2) show a dramatic difference on math (GSM8K) and coding (HumanEval, MBPP). MixtureVitae achieves scores of **0.53**, **0.32**, and **0.38**, respectively. This performance is considerably stronger than any other dataset, all of which remain near random performance on GSM8K (0.02-0.06) and cap at 0.13 on HumanEval and 0.22 on MBPP. Most notably, our base model outperforms the post-trained SmolLM2-1.7B-Instruct (Ben allal et al., 2025) model on GSM8K, HumanEval, and MBPP – despite the

Table 2: **Performance on math, code, and instruction-following tasks for 1.7B models.** We compare MixtureVitae—trained on a reasoning- and instruction-heavy, permissive-first mixture—against standard `open-sci-ref` baselines trained on predominantly web-based corpora. MixtureVitae shows a substantial lead in math and code tasks. Notably, the 1.7B MixtureVitae base model exceeds **SmolLM2-1.7B-Instruct** on GSM8K, HumanEval, and MBPP despite training on 300B rather than ≈11T tokens.

| Training Dataset | Tokens | IF-Eval | GSM8K | HumanEval | MBPP | Average |
|---|---|---|---|---|---|---|
| *Models Trained with `open-sci-ref` for 300B Tokens* | | | | | | |
| MixtureVitae | 300B | 0.19 | **0.53** | **0.32** | **0.38** | **0.36** |
| Comma-0.1 | 300B | 0.19 | 0.06 | 0.13 | 0.22 | 0.15 |
| CommonCorpus | 300B | 0.13 | 0.02 | 0.05 | 0.05 | 0.06 |
| C4 | 300B | 0.20 | 0.02 | 0.00 | 0.00 | 0.06 |
| SlimPajama | 300B | 0.14 | 0.02 | 0.05 | 0.00 | 0.05 |
| HPLT-2.0 | 300B | 0.17 | 0.02 | 0.00 | 0.00 | 0.05 |
| DCLM | 300B | 0.13 | 0.02 | 0.01 | 0.01 | 0.04 |
| Nemotron-CC-HQ | 300B | 0.09 | 0.03 | 0.02 | 0.00 | 0.03 |
| *Models Trained with `open-sci-ref` for 1T Tokens* | | | | | | |
| FineWeb-Edu | 1T | 0.20 | 0.03 | 0.00 | 0.00 | 0.06 |
| Nemotron-CC-HQ | 1T | 0.13 | 0.03 | 0.01 | 0.04 | 0.05 |
| DCLM | 1T | 0.15 | 0.03 | 0.00 | 0.01 | 0.05 |
| *Other Models* | | | | | | |
| SmolLM2-1.7B | 11T | 0.18 | 0.31 | 0.01 | 0.35 | 0.21 |
| SmolLM2-1.7B-Instruct | 11T | **0.28** | 0.37 | 0.28 | 0.37 | 0.33 |

latter being trained on ≈11T tokens (over 36× our budget). For scaling behavior on math/code/reasoning tasks with compute, see App. Sec. E.3; for evaluation on additional harder reasoning tasks, see App. Sec. E.4.

## 3.4 Test leakage and decontamination

To rule out test-set leakage as an alternative explanation for these gains, we perform a 13-gram exact-match decontamination sweep between MixtureVitae and all benchmarks (Appendix F). Document-level overlap is negligible for most tasks (e.g., at or below 0.0003% for ARC, HellaSwag, LAMBADA, OpenBookQA, and PIQA; see Table 10); contamination rates are modest for MMLU and BoolQ; for code benchmarks such as HumanEval and MBPP, contamination rates are higher but still small.

**Decontaminated Test Set Performance.** We re-evaluate all models on decontaminated test sets with all overlapping items removed. As shown in App. Tab. 13, the performance of MixtureVitae is consistent between the original and decontaminated versions. Crucially, the scores on GSM8K (0.54 decontaminated vs. 0.53 original) and MBPP (0.38 for both) remain stable, ruling out the possibility that our strong performance on math and coding is due to memorization of test items.

**Retraining on Decontaminated Shards**. To further alleviate concerns, we train a 1.7B model, removing the shards responsible for the majority of the contamination signal. As illustrated in Figure 4, removing these shards had no negative effect on downstream performance. The training trajectory of the decontaminated model tracks closely with the full MixtureVitae model, confirming that our results are not an artifact of dataset contamination.

## 3.5 Ablation Studies

To isolate the impact of primary data components in MixtureVitae, we define **Web** and **Instructions** subsets (see Figure 1; **Instructions** encompasses Reasoning & Instruction and Math parts of the full mixture) and conduct an ablation study on a 100B-token scale. We train three separate models: (1) MixtureVitae **(full)**, the complete dataset; (2) MixtureVitae **(w/o Web)**, removing the **Web** component; (3) MixtureVitae **(w/o Instructions)**, removing the **Instructions** component.

The average downstream performance of these models (Figure 5a) shows varying contributions by each component: The **Instructions** data is the most critical driver of performance, as its removal results in the

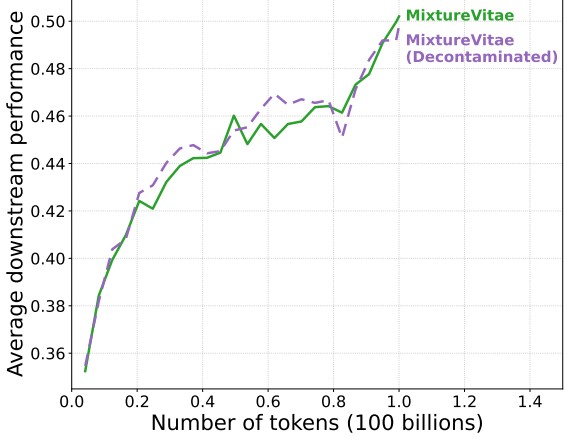
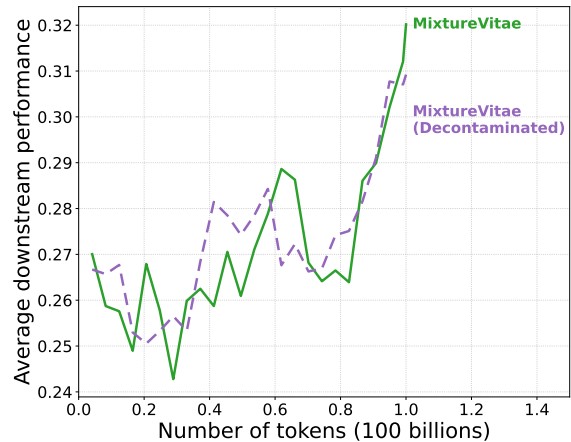

(a) Average accuracy across all tasks (as listed in Table 6) as a function of number of training steps.

(b) Accuracy on MMLU as a function of number of training steps.

Figure 4: **Validation of 1.7B model performance (100B token budget)**. The **MixtureVitae (Decontaminated)** model (purple, dashed), trained with dataset shards responsible for benchmark leakage removed, performs closely to the full **MixtureVitae** (green, solid) model. This confirms our results are not an artifact of test set leakage.

largest, consistent drop of average performance compared to other configurations. Removing **Instructions** particularly leads to severe drop on GSM8k (from 0.47 to 0.03) and MBPP, as shown in Figure 5b. Absent the **Instructions** data, the model fails to match the gains of the full mix, underscoring the essential role of instruction-following data in generalization.

Removing the **Web** component (**w/o Web**, blue dashed line) also results in a performance drop below the full dataset, albeit less dramatically. Figure 5b shows a drop from 0.47 to 0.41 on GSM8k, far less severe than the drop close to 0 for **w/o Instructions** and only slight changes on code evals. The comparison of ablation effects again highlights the **crucial role of instruction and reasoning data in achieving high performance**.

## 4   Related Work

LLM development is intrinsically linked to the scale and quality of pretraining datasets, which have become larger, more diverse, with a growing emphasis on provenance and licensing recently.

**Pioneering Large-Scale Datasets**. Early large-scale text corpora for language modeling often rely on web-crawled data for scale. C4 (Raffel et al., 2020), derived from Common Crawl, is instrumental in training the T5 model, setting standards for large-scale data cleaning and deduplication. Gao et al. (2020) then introduce The Pile, demonstrating the benefit of a more varied data mixture on model generalization and downstream performance. Similarly, ROOTS (Laurençon et al., 2022) supports the training of the BLOOM model with its 498 Common Crawl multilingual scrapes. While foundational, these datasets often have complex or unspecified licenses, mixing permissive data with content of unknown or non-commercial licensing, creating potential legal risks for commercial applications.

**Open and Reproducible Datasets.** Amidst many proprietary "black box" datasets, the community has pushed for more openness and reproducibility, moving toward permissive datasets that are also performant, e.g., RedPajama-1T (Weber et al., 2024) and its processing recipes (Touvron et al., 2023), Dolma (Soldaini et al., 2024) and its open-source curation toolkits, SILO (Min et al., 2024), FineWeb (Penedo et al., 2024) and DCLM (Li et al., 2024). Our work joins this effort, contributing a new risk-mitigated dataset featuring explicit consideration for the underlying copyright.

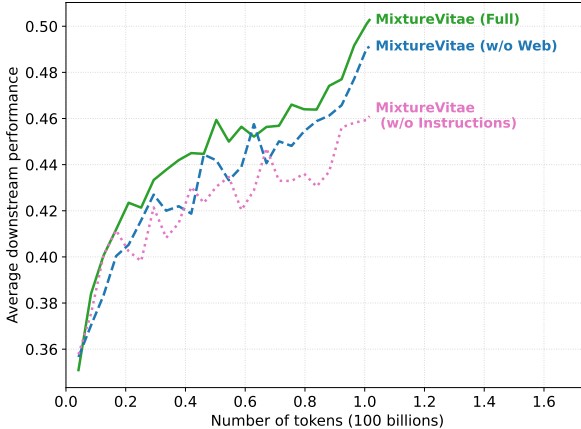

(b) A performance breakdown on math, coding and instruction following tasks for the ablated dataset variants. Best results are in **bold**. Numbers in red indicate strong performance drop.

| Training Dataset | IF-Eval | GSM8K | MBPP | Average |
|---|---|---|---|---|
| **MixtureVitae** | 0.14 | **0.47** | **0.34** | **0.25** |
| **MixtureVitae** (w/o Web) | 0.18 | 0.41 | 0.33 | **0.25** |
| **MixtureVitae** (w/o Instructions) | **0.19** | 0.03 | 0.14 | 0.14 |

(a) Ablation on full **MixtureVitae** against two versions, each excluding a data subset as indicated by `w/o`. Average performance on 11 downstream tasks.

Figure 5: An ablation study on components of the **MixtureVitae** dataset (100B token budget). Fig. 5a shows performance average on 10 downstream evals during training, while Fig. 5b shows scores on further separate math, code and instruction benchmarks which are not part of the average in (a). The evaluation setup is given in Table 7.

**Permissively Licensed and Synthetic Data.** Growing awareness of copyright and data ownership has spurred interest in datasets built solely from permissively licensed materials. The Stack (Kocetkov et al., 2023) curates such data for code-generation models, but creating a large, diverse, and high-quality corpus for natural language from exclusively permissive sources remains a challenge. Recent efforts like Common Corpus (Langlais et al., 2025) and The Common Pile (Kandpal et al., 2025) (the 8 TB data pool from which the Comma-0.1 training corpus is derived) advance the creation of large-scale corpora of permissively licensed and public-domain text. While foundational, our experiments (Section 3) show that models trained on them can lag in complex reasoning, math, and instruction following, suggesting that strictly permissive human text alone is insufficient to instill these advanced skills. **MixtureVitae** differs from Comma-0.1 in three ways: (i) in *unique content*, Comma-0.1's ≈1T training tokens comprise only ≈316B unique tokens due to up to 6× repetition of most sources (Table 7 of Kandpal et al. 2025), whereas **MixtureVitae** contains **422**B unique tokens; (ii) in *design*, The Common Pile avoids synthetic data, while **MixtureVitae** augments permissive sources with permissive-first synthetic reasoning/instruction data organized into explicit legal-risk tiers (Section 2.1.4); (iii) in *outcome*, **MixtureVitae** outperforms Comma-0.1 on language understanding (Table 1) and by a large margin on reasoning, math, and code (Table 2).

With this scarcity of high-quality reasoning and instruction data, researchers have turned to synthetic data. Alpaca (Taori et al., 2023) and OpenMathInstruct-1 (Toshniwal et al., 2024a) use instructional data for fine-tuning. Phi4 proposes using synthetic data for reasoning tasks (Abdin et al., 2024). Our work, **MixtureVitae**, extends these trends with a meticulously curated, permissive-first, risk-mitigated dataset augmented with targeted synthetic data, providing a strong, legally considered foundation for LLM training to mitigate copyright risks in many existing corpora.

While both our work and the concurrent Apertus project (Hernández-Cano et al., 2025) value openness and legal safety, they represent distinct, complementary design philosophies. First, regarding scale versus efficiency, Apertus optimizes for breadth, processing 15T tokens across 1800+ languages using retroactive filtering (e.g., `robots.txt`) on large web-scale datasets. In contrast, **MixtureVitae** focuses on data efficiency through a *positive inclusion strategy*, curating sources known to be permissive (e.g., government works, The Stack) and prioritizing English-centric reasoning density. Our results demonstrate that a reasoning-heavy mixture can achieve strong performance on MMLU, GSM8K, and MBPP with roughly 2% of the pretraining token budget of a dataset in the size range of Apertus. Finally, while Apertus releases recipes and reconstruction

scripts, MixtureVitae also provides a single, ready-to-use pretraining dataset, which strongly simplifies reproducibility and validation by other parties.

**Mixing Reasoning Data into Pre-Training.** Concurrent with our work, Akter et al. (2025) systematically investigate the "front-loading" of reasoning data, finding that injecting reasoning data into the pretraining phase establishes foundational capabilities that cannot be replicated by scaling supervised fine-tuning (SFT) alone. They observe an asymmetric principle where pretraining benefits most from the scale and diversity of reasoning patterns, while SFT relies more heavily on data quality. Similarly, Wang et al. (2025) augment pre-training text data with synthetically generated thinking trajectories. They observe that pre-training augmented with thinking traces strongly outperforms vanilla pretraining using matched compute and token budget (8B model, 100BT) on reasoning/math/language understanding evals. Our findings with MixtureVitae align with and extend this observation to the permissive dataset landscape: we show that by front-loading a diverse, risk-mitigated mixture of reasoning and instruction data, we can achieve competitive performance against non-permissive baselines even with a constrained token budget. For a dataset composition comparison of MixtureVitae to other permissive and non-permissive baselines, see Tab. 3.

## 5 Discussion & Conclusion

We have introduced MixtureVitae, a **permissive-first general-purpose pretraining corpus** that deliberately front-loads reasoning and instruction data, serving as proof-of-concept that permissively licensed and permissively-sourced real and synthetic data can achieve high performance comparable to non-permissive sources while simultaneously providing a strong reasoning-capable base model already before post-training. Our results suggest a shift in the **compliance–performance frontier**. MixtureVitae demonstrates that capabilities previously associated with mixed-license corpora are reachable with a permissive first, risk-mitigated approach. In our controlled 300B-token experiments, not only does MixtureVitae catch up to leading non-permissive baselines like DCLM and FineWeb-Edu, but our 1.7B base model also outperforms the *post-trained* SmolLM2-1.7B-Instruct–a model trained on ≈11T tokens–on GSM8K, HumanEval and MBPP.

**Mixing dominant fraction of reasoning & instruction data into pre-training.** MixtureVitae's performance is enhanced by the large proportion of reasoning and instruction data, as demonstrated in the ablation study in Section 3.5. Removing this subset ("w/o Instructions" in Fig. 5) causes a substantial degradation across tasks—far larger than the impact of removing the web component. This observation validates and extends the findings of Phi-4 (Abdin et al., 2024), showing that a permissive-first, risk-mitigated, and reasoning-heavy mixture can substitute vast quantities of generic web text, particularly under constrained token budgets. Importantly, while strongly boosting the performance on math/code tasks (Tab. 2), language understanding evals also stays strong, matching non-permissive baselines and outperforming other permissive datasets (Fig. 3, Tab. 1). A central finding is that MixtureVitae's overproportionally large fraction of reasoning, instruction, math, and code data–including synthetic data–does not come at the cost of general-purpose capabilities: language understanding evals remain strong, catching up with non-permissive baselines and outperforming other permissive datasets, while reasoning/math/code performance improves dramatically.

Beyond this specific corpus, the three-tier licensing scheme and its shard-level annotations provide a **concrete template for structuring risk-mitigated mixtures in future work**, and MixtureVitae as a whole serves as a reusable blueprint for compliant pretraining. We demonstrate a fully open, reproducible pipeline built on positive-inclusion "pseudo-crawling," tiered provenance tracking, targeted synthetic generation with audited seeds and decontamination controlling for test set leakage. As detailed in our scaling outlook (Appendix L), this recipe provides a path to extend compliant pretraining to the multi-trillion-token regime–via subset upsampling, multilingual expansion, and synthetic growth–providing the community with a sustainable alternative to the legal uncertainty of broad web scrapes.

## Acknowledgements

Huu Nguyen is thankful for and acknowledges discussions with Robert Kaczmarczyk on the ethical implications of training data; Colin Raffel on the inclusion of large scale instruction data in pretraining; Stella Biderman on various best practices for permissive datasets; Wojciech Kusa and the members of NASK – National

Research Institute on data safety; Veronika Laippala and Sampo Pyysalo of the University of Turku for their advice, and he especially thanks his wife Thao Tran for her enduring support.

Marianna Nezhurina, Harsh Raj, David Salinas and Jenia Jitsev acknowledge funding by EU Horizon under grant no. 101214398 (ELLIOT) and co-funding by EU from Digital Europe Programme under grant no. 101195233 (openEuroLLM). Marianna Nezhurina and Jenia Jitsev acknowledge co-funding from EU under Digital Europe Programme under grant no. 101198470 (LLMs4EU) and from EuroHPC Joint Undertaking programme under grant no. 101182737 (MINERVA), funding by the German Federal Ministry of Research, Technology and Space (BMFTR) under grant no. 01IS24085C (OPENHAFM), under the grant 01IS22094B (WestAI – AI Service Center West), and under the grant 16HPC117K (MINERVA).

We gratefully acknowledge the Gauss Centre for Supercomputing e.V. (GCS) for funding this work by providing computing time through the John von Neumann Institute for Computing (NIC) on the supercomputer JUWELS Booster and JUPITER at Jülich Supercomputing Centre (JSC), EuroHPC Joint Undertaking for computing time and storage on the EuroHPC supercomputer LEONARDO, hosted by CINECA (Italy) and the LEONARDO consortium through an EuroHPC Extreme Access grant EHPC-EXT-2023E02-068 and through EuroHPC AI Factory Large Scale Access grant EHPC-AIF-2025LS01-028, storage resources on JUST granted and operated by JSC and supported by Helmholtz Data Federation (HDF), computing time granted by the JARA and JSC on the supercomputer JURECA at JSC, computing time granted on prototype JEDI via JUREAP (JUPITER Early Access Program) grant at JSC and computing time granted via Gauss AI Competition on JUPITER through GCS and German Federal Ministry of Research, Technology and Space (BMFTR). LAION further acknowledges public storage grant by HuggingFace that allows us to provide convenient access to the output of the open-source research to broad community via HF repository.

We thank Robert Kaczmarczyk for coordination and support of EuroHPC Extreme Scale grant applications. Further thanks go for support provided by supercomputing facilities and their teams, especially to Damian Alvarez, Bjoern Hagemeier and Mathis Bode from Juelich Supercomputer Center (JSC, Germany) and to Laura Morselli from CINECA (Italy).

We also would like to thank all the members of the Ontocord[3], LAION[4], and Open-$\Psi$ (open-sci) communities[5] for providing a fertile ground for scientific exchange and open-source development.

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

# Appendix: MixtureVitae − Open Web-Scale Pretraining Dataset With High Quality Instruction and Reasoning Data Built from Permissive Text Sources

## A Reproducibility statement

We release our code and all further infos required for reproduction at `https://github.com/ontocord/mixturevitae`. Dataset is available at `https://huggingface.co/datasets/ontocord/MixtureVitae-v1` (decontaminated version `https://huggingface.co/datasets/ontocord/MixtureVitae-v1-decontaminated`)

### A.1 Dataset and Curation Recipes

- **Public Release:** The full **422B** token dataset, along with the 100B and 50B subsets used for scaling ablations experiments, will be made publicly available upon acceptance of this paper.

- **Curation Methodology:**

  - **Dataset Composition** The detailed list of sources and their composition are shown in Figure 6.
  - **Code**: We are including our data curation and math word problem generation scripts with the submission.

### A.2 Training Procedure

To ensure our experiments are directly comparable and reproducible, we adhered to a controlled, public framework.

- **Framework:** All experiments were conducted using the **open-sci-ref** training procedure (Nezhurina et al., 2025), which standardizes key factors affecting performance.

- **Architectures:** The exact model architectures for all four scales (0.13B, 0.4B, 1.3B, 1.7B) are detailed in Table 4.

- **Hyperparameters:** The complete training schedules and hyperparameters (learning rate, batch size, warmup, etc.) for both the 50B and 300B token budgets are specified in Table D.1.

- **Software:** Models were trained using Megatron-LM (Shoeybi et al., 2020) with the GPT-NeoX-20B tokenizer(Black et al., 2022).

- **Code**: We are including our training script with the submission.

### A.3 Evaluation and Analysis

Our evaluation protocol is fully specified to allow for independent verification of our results.

- **Framework:** All general and reasoning task evaluations were performed using the public LM Evaluation Harness (Gao et al., 2021).

- **Settings:** The exact settings for each benchmark, including the number of few-shot examples, are provided in Table 6 and Table 7.

- **Decontamination:** Our 13-gram decontamination protocol is detailed in Appendix F.

- **Code**: We are including our evaluation and decontamination scripts with the submission.

While model checkpoints and training logs are not included in the initial submission due to size and anonymity constraints, we plan to release these upon publication to facilitate future research.

# B   Limitations and Broader Impact Statement

**Limitations.**   While the dataset improves the current state-of-the-art in the legal risk mitigation of hgh-performing pretraining data, upstream provenance may still contain errors with respect to licensing. We mitigate by tiering sources, providing explicit shard-level audit metadata, and applying filtering and decontamination; we encourage downstream users to select tiers consistent with their risk posture. Further automation of licensing check procedures is a subject of future work. The dataset has a scale of 422B tokens, which is not sufficient for larger-scale pre-training, and future work should investigate scaling up the presented dataset composition recipe.

**Broader Impact Statement.**   The dataset improves transparency and reduces legal uncertainty for open pre-training, providing a safe ground for research, experimentation and development for the open-source community. It also can boost the trust of the general public into open-source machine-learning research that can be executed on well-validated, transparent artifacts with clear origins and widely accepted licensing schemes for broad re-use.

**Role of synthetic data and provenance-tagged tiering.**   A central design choice in **MixtureVitae** is the deliberate mixing of human-authored and synthetic data. Community evidence converges on such mixing providing a strong capability boost, as demonstrated in Phi-4 (Abdin et al., 2024), rephrasing-the-web (Maini et al., 2024), and Nemotron-CC (Su et al., 2025), along with further recent web-rephrasing and synthetic-augmentation efforts. At the same time, synthetic data introduces distinct quality-assurance questions compared to human-authored text: generator-model artifacts, distributional narrowness, and amplification of upstream biases may be difficult to detect with filtering techniques designed for scraped web data. **MixtureVitae**'s tiered, provenance-tagged design is specifically intended to make these decisions auditable: users can inspect, include, or exclude real and synthetic components at the shard level based on their quality and compliance posture.

**Dual-use risk of front-loading reasoning in base models.**   All base models - regardless of pretraining mixture - carry dual-use risks prior to alignment, and alignment remains the province of post-training (RLHF, DPO, etc.), which is out of scope for this study that focuses on pretraining base models. A distinct and currently open question is whether front-loading reasoning and instruction data into pretraining - as **MixtureVitae** does - changes the downstream alignment landscape: reasoning-capable base models may be harder to align in some respects on the one hand, but their stronger instruction-following scaffolding may also facilitate alignment on the other hand. Rigorously answering this question requires dedicated alignment experiments that we leave to future work. **MixtureVitae**'s tiered structure allows users who consider this risk salient to adjust the proportion of reasoning/instruction data included during pretraining.

# C   Dataset Composition and Comparison

This appendix provides a detailed view of the **MixtureVitae** corpus, both in relation to other datasets and in its internal construction.

**Shard Definitions and Mixing.**   It is important to note that the dataset shards and categories listed in this appendix serve as logical groupings for transparency, licensing audits, and ablation analysis. They do not dictate a rigid sequential training order. As noted in the main text, the physical construction of training batches utilizes domain-aware packing to maximize local coherence, prioritizing the density of reasoning and factual tokens over these high-level taxonomic boundaries.

Table 3 presents a high-level comparison of **MixtureVitae** against the other prominent pretraining datasets evaluated in our experiments, detailing their respective sizes, primary data types, and licensing philosophies.

Figure 6 presents the detailed composition of the **MixtureVitae** dataset. The individual components are color-coded by their primary dataset category, as presented in the main text.

Table 3: Comparison of large-scale pretraining datasets, grouped by their licensing philosophy to provide context for our performance results. **MixtureVitae** is unique in its combination of a risk-mitigated licensing approach and the inclusion of a large subset of reasoning, coding and instruction synthetic data.

| Dataset | Size (Tokens) | Primary Data Types | Licensing Philosophy |
|---|---|---|---|
| *Non-Permissive / Mixed-License Baselines* | | | |
| Nemotron-CC-HQ (Su et al., 2025) | $\approx$ 1.1T | Web, Synthetic | Unspecified |
| DCLM-baseline (Li et al., 2024) | $\approx$ 3.8T | Web, Code, Academic | Mixed / Unspecified |
| FineWeb-Edu (Penedo et al., 2024) | $\approx$ 1.3T | Web (Educational) | Unspecified |
| The Pile (Gao et al., 2020) | $\approx$ 183.28B | Web, Books, Code | Mixed / Unspecified |
| SlimPajama (Shen et al., 2024) | $\approx$ 627B | Web, Books, Code | Mixed / Unspecified |
| C4 (Raffel et al., 2020) | $\approx$ 156B | Web | ODC-BY |
| HPLT-2.0 (eng.) (Burchell et al., 2025) | $\approx$ 2.86T | Web, Books, News | Mixed / Unspecified |
| *Permissive Baselines* | | | |
| CommonCorpus (Langlais et al., 2025) | $\approx$ 2T | Web, Curated | Strictly Permissive |
| Comma-0.1 (Kandpal et al., 2025) | $\approx$ 1T | Web, Curated | Strictly Permissive |
| KL3M (Bommarito et al., 2025) | $\approx$ 580B | Web, Curated | Strictly Permissive |
| OLC (Min et al., 2024) | $\approx$ 228B | Web, Curated | Strictly Permissive |
| *Our Contribution* | | | |
| **MixtureVitae** | $\approx$ **422B** | **Web, Curated, Synthetic** | **Permissive-First, Risk-Mitigated** |

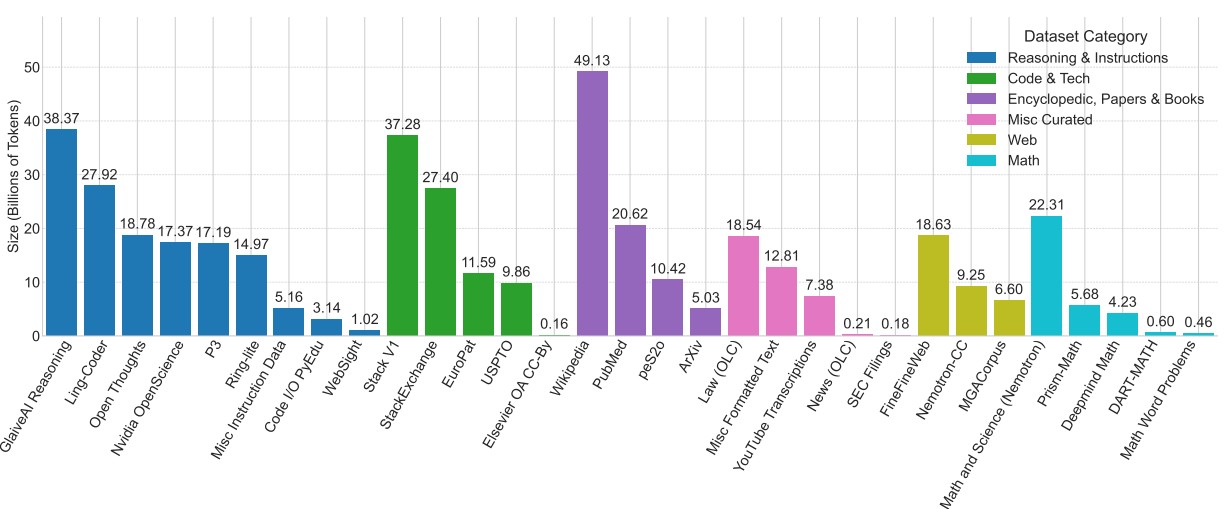

Figure 6: Detailed composition of the **MixtureVitae** dataset.

- **Code & Tech (Blue):** This domain is anchored by our largest code sources, Stack V1 and Ling-Coder, and supplemented by StackExchange.

- **Reasoning & Instruction (Green):** The largest contributor to this category is Open Thoughts , followed by P3 and NVIDIA OpenScience.

- **Encyclopedic, Papers & Books (Purple):** This category is dominated by Wikipedia, the single largest component in the dataset. It is complemented by large-scale text from PubMed and arXiv.

- **Math (Cyan):** The math component is a diverse mixture of sources, led by the Math and Science (Nemotron) corpus and Prism-Math.

- **Web (Yellow):** Our web data is primarily sourced from corpora such as SEC Filings, MGACorpus, and FineFineWeb.

- **Misc Curated (Pink):** This category includes a variety of high-quality curated sources, notably Law (Open License Corpus) and YouTube Transcriptions.

# D   Experiment Setup Details

To ensure full reproducibility, this appendix details the complete experimental setup. This includes the model architectures for all scales, the training hyperparameters for both 50B and 300B token budgets, and the specific settings used for all general evaluation benchmarks.

Table 4: **open-sci-ref** (Nezhurina et al., 2025) model architecture and scales. We used tied embedding weights in all experiments.

| Parameters (B) (Non-Emb + Emb) | Layers | Hidden | Heads | FFN Hidden | Memory | FLOPs |
|---|---|---|---|---|---|---|
| $0.1 + 0.03 = 0.13$ | 22 | 512 | 8 | 2256 | 0.89 GB | $7.8 \times 10^8$ |
| $0.35 + 0.05 = 0.40$ | 22 | 1024 | 16 | 3840 | 2.88 GB | $2.4 \times 10^9$ |
| $1.21 + 0.10 = 1.31$ | 24 | 2048 | 32 | 5440 | 7.544 GB | $7.9 \times 10^9$ |
| $1.61 + 0.10 = 1.71$ | 24 | 2048 | 32 | 8192 | 9.884 GB | $1.0 \times 10^{10}$ |

Table 5: The training schedules used in our experiments.

| Tokens | Global Batch Size (tokens) | Iterations | Learning Rate | Warmup | Cooldown (20%) |
|---|---|---|---|---|---|
| 50B | 4.12M | 11,921 | $4 \times 10^{-3}$ | 1,000 | 2,384 |
| 300B | 4.12M | 72,661 | $4 \times 10^{-3}$ | 25,000 | 14,532 |

## D.1   Training Setup Parameters

This appendix details the exact model architectures and training hyperparameters used for all experiments, ensuring full reproducibility.

We adopt the standard architectures and scales from the **open-sci-ref** framework to allow for a fair and direct comparison against other published baselines. All models were trained with tied embedding weights.

**Model Architecture**  Table 4 defines the four model scales used in our study. The columns are defined as follows:

**Parameters (B) (Non-Emb + Emb)** The total model parameters in billions, separated into **Non-Embedding** (Non-Emb) parameters (the core transformer blocks) and **Embedding** (Emb) parameters (the token lookup tables). As noted in the caption, we used tied embedding weights.

**Layers**  The total number of transformer blocks stacked in the model.

**Hidden**  The hidden size (or embedding dimension, $d_{\mathrm{model}}$) of the model.

**Heads**  The number of attention heads in the multi-head attention mechanism.

**FFN Hidden**  The inner dimension of the Feed-Forward Network (FFN) layer within each transformer block.

**Memory**  The approximate VRAM required to store the model weights, in bfloat16.

**FLOPs**  An approximation of the training compute cost using the **6N** rule: a standard estimate for a transformer's forward-and-backward pass, where **N** is the number of *non-embedding* parameters (Kaplan et al., 2020).

**Training Schedules**  Table 5 defines the training hyperparameters for our two main experimental runs (50B and 300B tokens). We use a single stage training with no post-training.

**Tokens**  The total number of tokens in the training run.

**Global Batch Size (tokens)** The total number of tokens processed in a single training step (i.e., one gradient update) across all GPUs.

**Iterations**  The total number of training steps.

**Learning Rate**  The peak learning rate used during training.

**Warmup**  The number of initial *iterations* (steps) over which the learning rate linearly increases from 0 to its peak value.

**Cooldown (20%)**  The number of final *iterations* (the last 20% of training) over which the learning rate decays to zero.

### D.2  Evaluation Settings

We used the `lm-evaluation-harness` (Gao et al., 2021) for all general evaluations. The specific tasks and few-shot counts are detailed in Table 6. The settings for the reasoning tasks (e.g., GSM8K, IFEval) are listed separately in Table7.

## E  Additional Experiments

This appendix provides additional experimental results to supplement the findings presented in the main paper. We offer a more granular breakdown of the 300B token experiment, analyze performance at a smaller 50B token scale to assess the generalization of our results, and report the results of a model red-teaming analysis to evaluate the model's safety profile.

### E.1  300B Experiment - Detailed Results

The detailed results for each evaluated task (contributing to the average over 10 tasks as shown in Figure 3) are given in Figure 7. Despite its substantial proportion of instruction and reasoning data which gives MixtureVitae exceptional performance for base model of ts scale on reasoning related tasks, MixtureVitae demonstrates also strong performance on language tasks that are typically associated with pretraining on broad web scrapes (see also Table 1 in main results Sec. 3).

Table 6: General evaluation benchmark settings. All tasks use Accuracy as the primary metric.

| Task | Citation | # of Shots |
|------|----------|------------|
| MMLU | Hendrycks et al. (2021a) | 5 |
| HellaSwag | Zellers et al. (2019) | 10 |
| CommonSenseQA | Talmor et al. (2019) | 10 |
| ARC-Challenge | Clark et al. (2018) | 10 |
| ARC-Easy | Clark et al. (2018) | 10 |
| PIQA | Bisk et al. (2020) | 10 |
| BoolQ | Clark et al. (2019) | 10 |
| Winogrande | Sakaguchi et al. (2021) | 0 |
| OpenBookQA | Mihaylov et al. (2018) | 0 |
| COPA | Roemmele et al. (2011) | 0 |
| LAMBADA | Paperno et al. (2016) | 0 |

Table 7: Evaluation settings for reasoning tasks. All tasks use Accuracy as the primary metric. To execute the evaluation, we used LM Evaluation Harness Gao et al. (2021).

| Task | Citation | # of Shots |
|------|----------|------------|
| GSM8k | Cobbe et al. (2021) | 4 |
| IFEval | Zhou et al. (2023) | 0 |
| MBPP | Austin et al. (2021) | 4 |
| GPQADiamond | Rein et al. (2024) | 0 |
| MATH500 | Lightman et al. (2023) | 0 |
| LiveCodeBench | Jain et al. (2024a) | 0 |

### E.2   Performance at 50B Tokens Scale.

To assess performance on a smaller reference tokens scale, we also evaluated models trained on a 50B token subset of each dataset. The results, shown in Figure 8 and Figure 9, indicate that the advantages of **MixtureVitae** manifest already at the smaller token scales. Figure 8 shows that **MixtureVitae** establishes a consistent performance lead over other permissive datasets within the first 50B tokens, especially at the 1.3B and 1.7B model scales. The per-benchmark analysis further reinforces this finding (see Figure 9). On MMLU, **MixtureVitae** is the only permissive dataset to show a significant learning signal early in training, demonstrating that its composition provides immediate benefits, which might be both due to knowledge rich and instruction like content. Arguably, this suggests that the reasoning capability shown by **MixtureVitae** is not a late-stage phenomenon but rather an indication of efficient instillation from the early stages of training. This strong initial performance underscores the learning efficiency of **MixtureVitae**, making it a compelling choice for achieving high performance with less computational cost.

### E.3   Comparison via Compute Scaling Behavior

Figure 10 provides an overview across both compute and model scales, extending the comparison at the single 1.7B model scale shown in Figure 3 and Table 1 of the main paper to smaller model scales while keeping the token budget consistently fixed at 300B. We plot average downstream performance over the same 11 tasks as a function of training FLOPs. The left panel covers the controlled `open-sci-ref` setup, where all reference datasets and **MixtureVitae** are trained at the same 300B-token budget across the 0.13B / 0.4B / 1.3B / 1.7B model scales; the right panel adds published external models (SmolLM2, Qwen2.5, DCLM) that span a wider compute range and use various training recipes and model architectures.

In the controlled comparison (left panel), **MixtureVitae** exhibits a scaling trend on par with the strong non-permissive baselines. At the smallest model scales the gap to the top mixed-license corpora (FineWeb-Edu, DCLM, Nemotron-CC-HQ) is substantial, but it narrows steadily as compute grows: by the 1.7B

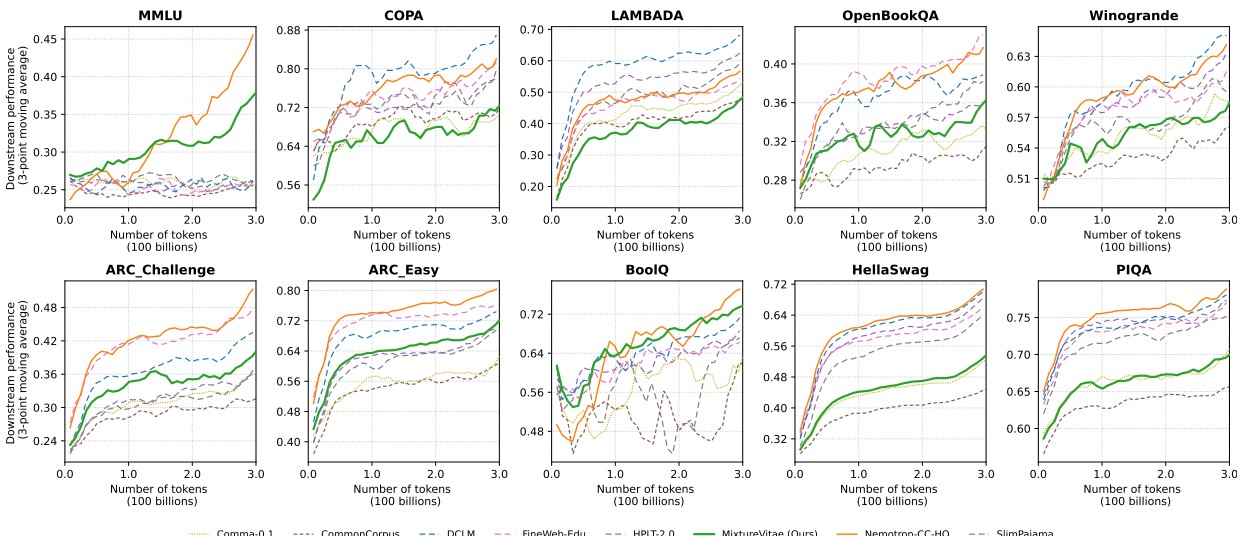

Figure 7: Comparing performance of 1.7B models trained on **MixtureVitae** and baseline datasets for a 300B token budget. While some evaluations provide clear dataset rankings (e.g. ARC, Hellaswag, Lambda), others do not provide a good signal for dataset comparison, on an individual basis.

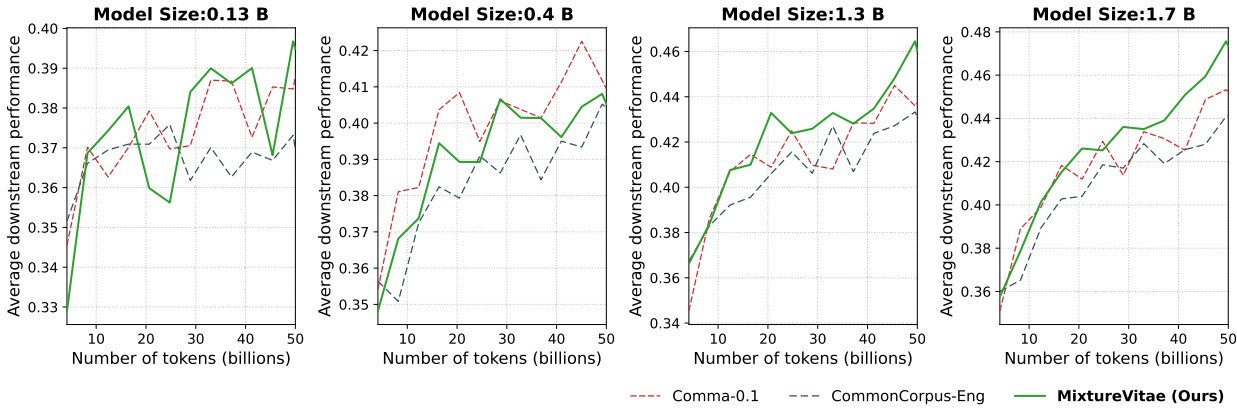

Figure 8: Average performance of permissive datasets after 50B training tokens on various model scales. **MixtureVitae** shows an early and consistent lead at larger model scales.

/ 300B operating point **MixtureVitae** clearly outperforms every other permissive corpus (Comma-0.1, CommonCorpus) and catches up with FineWeb-Edu and DCLM, while only Nemotron-CC-HQ retains a visible margin. Compared to published external models (right panel), **MixtureVitae** sits on a favorable scaling trajectory, tracing a steeper compute-to-performance trend than SmolLM2 or Qwen2.5 across the overlapping compute range. Together with the 50B-token analysis above, this indicates that the advantage of **MixtureVitae** is not a single-point artifact but a consistent property of its scaling behavior, with the permissive–non-permissive gap shrinking as compute increases.

To complement the scaling overview on language evaluations, Figure 11 reports the same compute-vs-performance overview for the reasoning, math, and code tasks (average over GSM8K, IFEval, MBPP – the same three benchmarks used in Table 2 and Figure 5b of the main paper). The two panels follow the same conventions as Figure 10.

The math/code/reasoning scaling clearly reveals the advantages of front-loading math/code/instruction data in **MixtureVitae**. In the controlled comparison (left panel), **MixtureVitae** shows a strong scaling

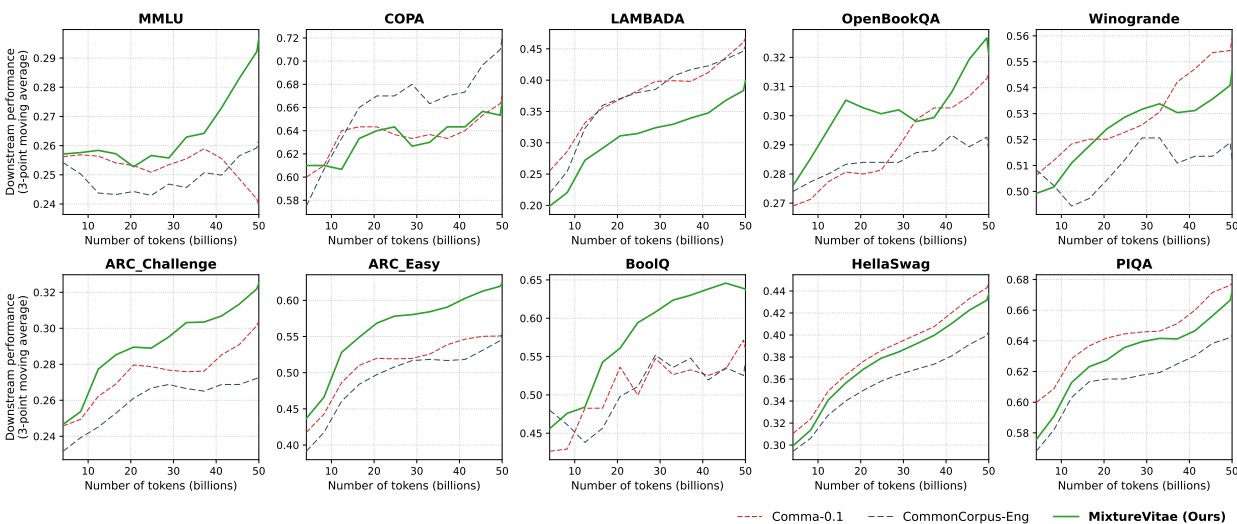

Figure 9: Per-benchmark performance of permissive datasets after 50B training tokens on 1.7B model scale. **MixtureVitae**'s advantage on MMLU is apparent even at this early stage.

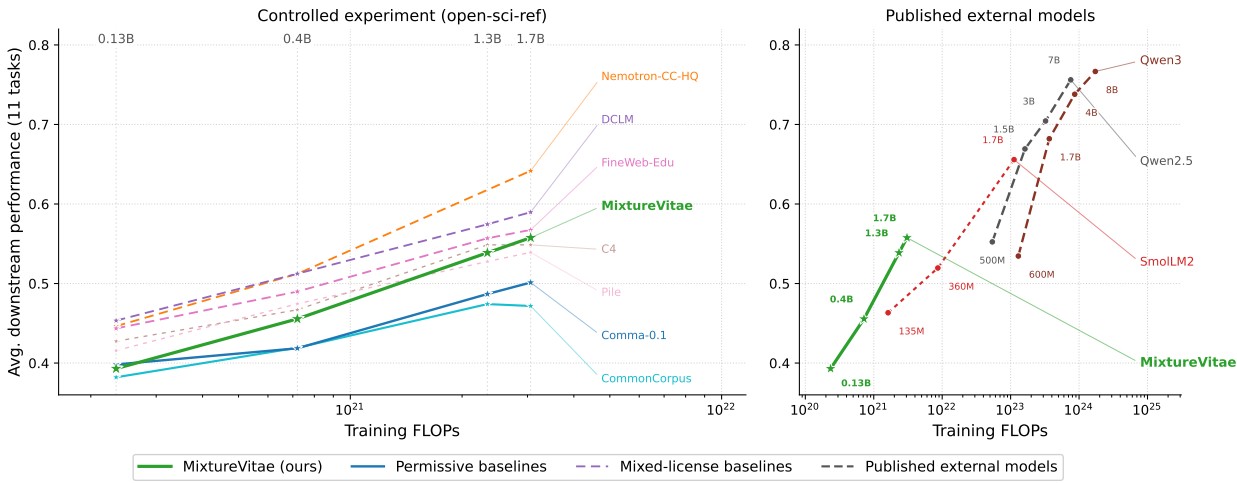

Figure 10: Average downstream performance on the 11-task suite (same tasks as Figure 3 and Table 1) as a function of training FLOPs. **Left:** controlled `open-sci-ref` experiment at 300B tokens across model scales 0.13B–1.7B for **MixtureVitae**, permissive baselines (Comma-0.1, CommonCorpus), and mixed-license baselines (Pile, C4, FineWeb-Edu, DCLM, Nemotron-CC-HQ). **Right: MixtureVitae** placed alongside published external models (SmolLM2 135M/360M/1.7B, Qwen2.5 0.5B/1.5B/3B/7B, Qwen3 0.6B/1.7B/4B/8B).

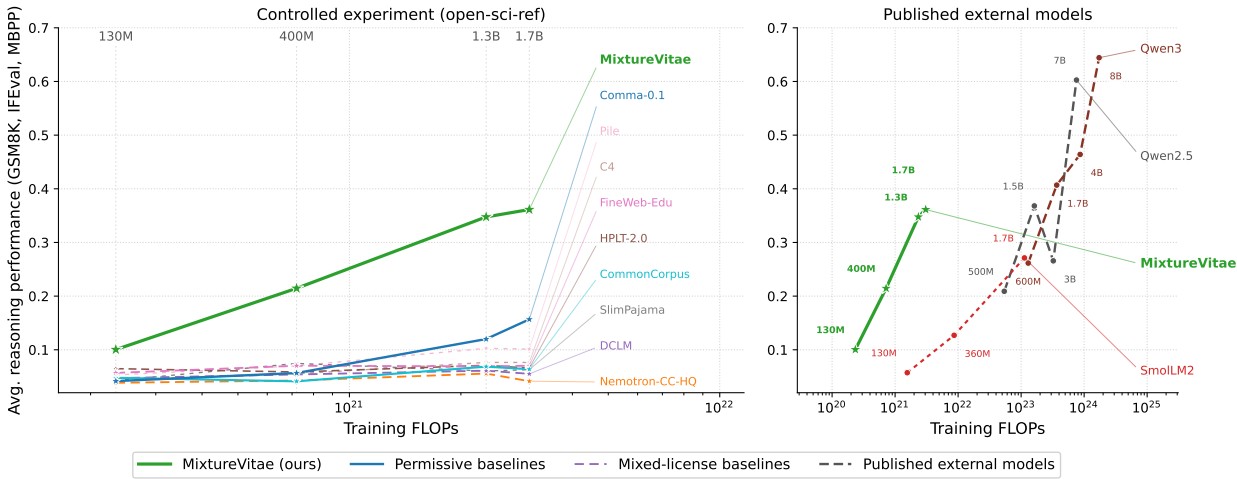

Figure 11: Average performance on math/code/reasoning (GSM8K, IFEval, MBPP – same three tasks as Table 2 and Figure 5b) as a function of training FLOPs. **Left:** controlled `open-sci-ref` experiment at 300B tokens across model scales 0.13B–1.7B for **MixtureVitae**, permissive baselines (Comma-0.1, CommonCorpus), and mixed-license baselines (Pile, C4, FineWeb-Edu, HPLT-2.0, SlimPajama, DCLM, Nemotron-CC-HQ). **Right:** **MixtureVitae** placed alongside published external models (SmolLM2 135M/360M/1.7B, Qwen2.5 0.5B/1.5B/3B/7B, Qwen3 0.6B/1.7B/4B/8B).

trend on math, code, and instruction-following that leaves every other corpus – both permissive and non-permissive – far behind, with the gap widening as model scale grows: at 1.7B / 300B, **MixtureVitae** reaches average performance above 0.35 while every baseline, including the strongest non-permissive ones (DCLM, Nemotron-CC-HQ, FineWeb-Edu), remains below 0.16. Compared to published external models (right panel), **MixtureVitae** clearly leaves SmolLM2 behind and matches or surpasses the Qwen2.5 and Qwen3 model families at matched compute, despite the Qwen series using substantially larger pretraining token budgets and multi-stage training recipes. Together, Figures 10 and 11 indicate that **MixtureVitae**'s reasoning-heavy permissive composition translates into a particularly favorable compute scaling behavior on math, code, and instruction-following, while keeping language understanding performance on par with the strongest non-permissive baselines.

### E.4 Evaluation on Harder Reasoning Benchmarks

To probe the signal of harder reasoning evaluations beyond the DCLM core benchmark set used in the main paper, we additionally evaluate our 1.7B `open-sci-ref` models on three more recent and challenging benchmarks: **MATH500** (Hendrycks et al., 2021b), **GPQA Diamond** (Rein et al., 2023), and **LiveCodeBench** (Jain et al., 2024b) (pass@1, $n_{\text{repeats}} = 3$). Results are reported in Table 8.

These results confirm the expectation discussed in the main text: benchmarks targeted at larger or post-trained models yield little to no signal for 1.7B base models. All `open-sci-ref` baselines–both permissive and non-permissive–score 0.00 on MATH500 and LiveCodeBench, and GPQA Diamond scores cluster at or below random chance (0.25 for 4-way multiple choice) for every base model. Notably, **MixtureVitae** is the *only* pretraining corpus at this scale to produce non-zero signal on MATH500 (0.18) and LiveCodeBench (0.13), matching the post-trained SmolLM2-1.7B-Instruct on MATH500 and more than doubling it on LiveCodeBench, despite being a single-stage base model trained on ≈36× fewer tokens.

### E.5 Model Red Teaming

To evaluate the safety of the model trained on **MixtureVitae** for 300B tokens, we performed a red-teaming analysis to measure the Attack Success Rate (ASR) against three standard benchmarks: **ToxiGen** (Hartvigsen

Table 8: **Results on harder reasoning benchmarks for 1.7B models**. *Italic* marks best among permissive datasets; **bold** marks best overall. GPQA-D has a random baseline of 0.25 for 4-way multiple choice; values marked with * are at or below random chance. **MixtureVitae** is the only pretraining corpus at this scale to produce non-zero signal on MATH500 and LiveCodeBench, matching or exceeding the post-trained SmolLM2-1.7B-Instruct (trained on ≈36× more tokens). Unlike the evaluations in the main paper, these additional benchmarks were *not* explicitly decontaminated, so scores should be interpreted with appropriate caution.

| Training Dataset | License | Tokens | Training | MATH500 | GPQA-D | LCB |
|---|---|---|---|---|---|---|
| *open-sci-ref 1.7B, 300B tokens, single-stage pretraining* | | | | | | |
| MixtureVitae | permissive | 300B | single-stage | *0.18* | 0.25* | *0.13* |
| Comma-0.1 | permissive | 300B | single-stage | 0.00 | 0.23* | 0.00 |
| CommonCorpus | permissive | 300B | single-stage | 0.00 | 0.24* | 0.00 |
| C4 | mixed | 300B | single-stage | 0.00 | 0.21* | 0.00 |
| FineWeb-Edu | mixed | 300B | single-stage | 0.00 | 0.21* | 0.00 |
| SlimPajama | mixed | 300B | single-stage | 0.00 | 0.22* | 0.00 |
| HPLT-2.0 | mixed | 300B | single-stage | 0.00 | 0.20* | 0.00 |
| *Other models* | | | | | | |
| SmolLM2-1.7B | mixed | 11T | multi-stage pretraining | 0.00 | 0.22* | 0.00 |
| SmolLM2-1.7B-Instruct | mixed | 11T | multi-stage pretraining + post-training | 0.17 | **0.29** | 0.05 |

et al., 2022), **Do-Not-Answer** (Wang et al., 2024), and **AdvBench** (Zou et al., 2023). The results (Table 9) shows that our model is competitive with the baselines.

The model responses were evaluated using two safety classifiers: (i) **Llama Guard-8B** (Inan et al., 2023), used to evaluate the **Do-Not-Answer** and **AdvBench** datasets, while (ii) the **toxigen_roberta** classifier (Logacheva et al., 2022) was used for the **ToxiGen** benchmark.

Table 9: Attack Success Rate in %, lower is better. All models are trained with the same **open-sci-ref** procedure (300B-token budget) while varying only the pretraining dataset.

| Benchmark | MixtureVitae | Comma | CommonCorpus-Eng | Nemotron-HQ-CC |
|---|---|---|---|---|
| ToxiGen | 8.07 | 9.04 | 12.77 | 10.21 |
| Do-Not-Answer | 28.22 | 24.71 | 21.62 | 20.98 |
| AdvBench | 86.92 | 92.12 | 70.58 | 85.77 |

# F   Contamination Analysis

## F.1   Contamination Detection Protocol

To ensure the integrity of our evaluation, we implemented a comprehensive decontamination protocol to measure the overlap between our training dataset and all evaluation benchmarks we report results on. This protocol consists of three main stages: Index Construction, Dataset Scanning, and Leakage Reporting.

### F.1.1   Index Construction

The first stage creates a compact, indexed set of unique n-grams from all benchmark evaluation data.

1. **Text Normalization:** All text from the benchmarks is processed through a normalization pipeline, similar to Laurençon et al. (2022): (1) Unicode normalization (NFKC), (2) conversion to lowercase, (3) tokenization, and (4) removal of a predefined list of common English stop words. This procedure focuses the resulting n-grams on substantive content.

2. **N-gramming and Filtering:** We generate 13-grams, a common n-gram size for this task Brown et al. (2020); Gao et al. (2020) from the normalized token lists. As in Laurençon et al. (2022), a set

of regular expressions is used to filter out common boilerplate, exam instructions, and formatting artifacts.

3. **Train/Test De-duplication:** as in Gao et al. (2020), we compute the set of all 13-gram hashes from the `train` split and subtract this set from the 13-gram hashes generated from the `test` split. This ensures our index only contains n-grams that are unique to the evaluation set.

### F.1.2  Dataset Scanning

The second stage analyzes the target training dataset against the generated index.

1. **Document Processing:** Each document in the training dataset is processed using the *exact same* normalization, 13-gramming, and hashing pipeline used for index construction.

2. **Contamination Criteria:** A document is flagged as "contaminated" if it meets two criteria, based on the set intersection of its n-gram hashes with the benchmark index:

   - **Minimum Hits:** The number of distinct matching n-grams is $\geq 3$.
   - **Minimum Coverage:** As proposed in Rae et al. (2022), the coverage of matching n-grams is $\geq 0.1\%$. Coverage is defined as:

$$\text{Coverage} = \frac{\text{distinct\_hits}}{\text{total\_unique\_13grams\_in\_doc}}$$

### F.1.3  Leakage Reporting

The final stage aggregates the scan results into a summary report.

1. **Numerator (Leaked N-grams):** The procedure aggregates the reports from all scanned partitions. It performs a global *set union* to find all unique n-gram hashes that were found *at least once* in the target dataset, aggregated by benchmark source. This provides the unique\_ngrams\_leaked count for each benchmark.

2. **Denominator (Total N-grams):** The procedure retrieves the pre-computed metadata to obtain the total unique n-gram count for each benchmark.

3. **Final Metric:** As proposed in Touvron et al. (2023), the **Leak Percentage** for each benchmark is then calculated as:

$$\text{Leak Percentage} = \frac{\text{unique\_ngrams\_leaked}_{\text{benchmark}}}{\text{total\_unique\_ngrams\_in\_index}_{\text{benchmark}}} \times 100$$

### F.2  Contamination Report

We executed our 13-gram contamination scan across the entire $345\,697\,271$ documents of the **MixtureVitae** dataset. The global summary of contaminated documents per benchmark is presented in Table 10.

The results confirm that for the vast majority of benchmarks—including ARC, HellaSwag, LAMBADA, OpenBookQA, and PIQA—the document-level contamination rate is negligible (at or below 0.0003%), strongly validating the integrity of our evaluation on these tasks.

The scan did, however, flag a minor overlap for MMLU (0.0098%) and BoolQ (0.0087%), and a more significant overlap for our key code benchmarks: HumanEval (0.0988%) and MBPP (0.0878%). This overlap in code benchmarks is a known challenge when including large-scale permissive code corpora like The Stack, which may naturally contain snippets of common coding problems (a "source overlap" rather than a direct "test-set leak").

Table 10: Global contamination summary by document count, based on a 13-gram overlap scan. This table shows the total number of documents in **MixtureVitae** that contained at least one overlapping n-gram from each benchmark's test set. The total documents in **MixtureVitae** is 345 697 271 and the overall contamination rate is 0.1420%.

| Benchmark | Contaminated Docs | Contamination Rate (%) |
|---|---|---|
| ALERT | 12 | 0.0000% |
| ARC | 17 | 0.0000% |
| BoolQ | 30 144 | 0.0087% |
| CommonSenseQA | 0 | 0.0000% |
| GPQA | 1077 | 0.0003% |
| GSM8K | 230 | 0.0001% |
| HellaSwag | 186 | 0.0001% |
| HumanEval | 341 554 | 0.0988% |
| IfEval | 756 | 0.0002% |
| LAMBADA | 23 | 0.0000% |
| MBPP | 303 558 | 0.0878% |
| MMLU | 33 922 | 0.0098% |
| OpenBookQA | 60 | 0.0000% |
| PIQA | 5 | 0.0000% |
| SimpleQA | 98 | 0.0000% |

Table 11: Benchmark test set sizes (number of examples) for the original benchmarks versus the final decontaminated versions. The 'Decontaminated' column shows the reduced set size after removing all examples with detected 13-gram training data overlap.

| Dataset | Original | Decontaminated |
|---|---|---|
| MBPP | 500 | 331 |
| IFEval | 541 | 429 |
| GSM8K | 1319 | 1235 |
| MBPP+ | 378 | 339 |

## F.3 Full Decontamination Experiment

We show here in detail to complement Sec. 3.4 the evaluation comparing training on original and decontaminated dataset version (Tab. 13). We observe no significant difference between the both.

In addition to the decontamination experiments detailed in Section 3.4, we also performed an experiment in which we removed *every* document in **MixtureVitae** that was flagged as contaminated by our 13-gram procedure (Appendix F.2) and retrained a 1.7B model for 300B tokens under the `open-sci-ref` setup. The results–shown in Figure 12–indicate that the fully decontaminated variant performs slightly *better* than the

Table 12: Contamination sources for the **MMLU** benchmark in **MixtureVitae**, sorted by the number of contaminated documents, high to low.

| Dataset Shard | Contaminated Docs |
|---|---|
| Misc-Instruct | 14 649 |
| DART-Math (Tong et al., 2024) | 11 102 |
| Nemotron Science & Math (Bercovich et al., 2025) | 4793 |
| MGACorpus (Hao et al., 2025) | 241 |
| (All Remaining) | 3137 |

Table 13: Validating math, code, and instruction performance by comparing original (Orig) vs. decontaminated (Decont) test sets for 1.7B models trained for 300B tokens. MixtureVitae's high scores are shown to be genuine, as performance is maintained after removing all overlapping test items. This confirms the model's capabilities are not an artifact of test set leakage.

| Training Dataset | GSM8K | | GSM8K-CoT | | MBPP | | MBPP+ | | IFEval | |
|---|---|---|---|---|---|---|---|---|---|---|
| | Orig | Decont | Orig | Decont | Orig | Decont | Orig | Decont | Orig | Decont |
| **MixtureVitae** | 0.53 | 0.54 | 0.50 | 0.50 | 0.38 | 0.38 | 0.55 | 0.59 | 0.19 | 0.23 |
| SmolLM2 | 0.30 | 0.30 | 0.28 | 0.29 | 0.35 | 0.35 | 0.48 | 0.48 | 0.17 | 0.20 |
| Comma-0.1 | 0.06 | 0.06 | 0.09 | 0.09 | 0.21 | 0.23 | 0.28 | 0.28 | 0.18 | 0.20 |
| CommonCorpus | 0.02 | 0.01 | 0.01 | 0.01 | 0.02 | 0.02 | 0.04 | 0.05 | 0.12 | 0.16 |
| C4 | 0.01 | 0.01 | 0.01 | 0.02 | 0.00 | 0.00 | 0.00 | 0.00 | 0.20 | 0.21 |
| DCLM | 0.01 | 0.02 | 0.02 | 0.02 | 0.01 | 0.00 | 0.02 | 0.02 | 0.12 | 0.13 |
| FineWeb | 0.02 | 0.01 | 0.03 | 0.03 | 0.00 | 0.00 | 0.00 | 0.00 | 0.18 | 0.20 |
| HPLT | 0.02 | 0.02 | 0.02 | 0.02 | 0.00 | 0.00 | 0.00 | 0.00 | 0.17 | 0.21 |
| Nemotron-CC-HQ | 0.03 | 0.02 | 0.03 | 0.03 | 0.00 | 0.00 | 0.00 | 0.00 | 0.09 | 0.10 |
| SlimPajama | 0.02 | 0.02 | 0.02 | 0.02 | 0.00 | 0.00 | 0.00 | 0.00 | 0.14 | 0.15 |

original MixtureVitae model, further addressing concerns that our benchmark results might be inflated by data leakage.

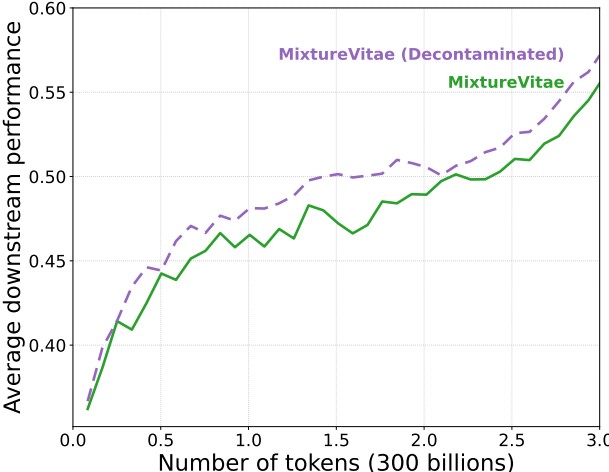

Figure 12: **1.7B model performance on a fully decontaminated dataset**. The model trained on the fully decontaminated MixtureVitae corpus (purple, dashed) performs slightly better than the model trained on the full MixtureVitae dataset (green, solid), further indicating that benchmark gains are not driven by contaminated examples.

## F.4 Discussion on Decontamination Methodology

Our decontamination pipeline employs the standard 13-gram exact-match procedure (Abdin et al., 2024) to ensure high precision, scalability and comparability with prior baselines. While we acknowledge that exact matching overlooks paraphrased content, we avoided approximate methods (e.g., LSH, embedding-based) due to their tendency to produce false positives on common factual or algorithmic templates (Lee et al., 2022). As noted in Penedo et al. (2024), aggressive removal of semantically similar content risks distorting the training distribution by discarding high-value instructional data.

# G   Synthetic Math Data Generation

The synthetic math dataset was programmatically generated to produce a diverse range of mathematical problems and their solutions. The generation process covers a wide array of mathematical domains, including fundamental arithmetic operations, multi-term fractional expressions, and the step-by-step solution of algebraic linear equations. A key component of the dataset consists of word problems, where numerical challenges are embedded in narrative scenarios.

A significant feature of this generation pipeline is the creation of detailed, step-by-step solutions formatted as a chain-of-thought. For many problem categories, the scripts produce a human-readable explanation of the entire solution process. This is achieved by using a variety of randomized natural language templates to describe each logical step, such as carrying a digit in addition or isolating a variable in an equation.

Following the initial generation, a final post-processing step is applied to format the dataset for model training. This stage programmatically identifies data entries containing human-like, explanatory text by searching for common instructional words. For these selected entries, a descriptive header (e.g., "Here are examples of addition, division exercises") is dynamically generated. The content and phrasing of this header are randomized and based on the mathematical operations present within the text, adding significant linguistic diversity.

For example, in the generated math problem below, a model may be able to generalize to new numbers, but if the problem were to add three students instead of two, the model may not be robust enough to generalize. We leave this analysis for future work.

```
The age difference between Sarah and Asaf's age is half the
total number of pencils Sarah has. The sum of their ages is
132, and Sarah is 27 years old. If Asaf has 60 more pencils than
Sarah, calculate the total number of pencils they have together.
Solution: If the sum of their ages is 132, and Sarah is 27 years
old, Asaf is 105 years old.
The age difference between Sarah
and Asaf's age is 105-27 = 78.
Since the age difference between Sarah and Asaf's age is half
the total number of pencils Sarah has, Sarah has 2*78 = 156 pencils.
If Asaf has 60 more pencils than Sarah, Asaf has 156+60= 216 pencils.
Together, they have 156 + 216 = 372 pencils.
```

# H   Our Position on Using Governmental and Other Works Under Fair Use and Related Ethical and Legal Basis

To contextualize our licensing tiers and clarify the rationale behind including certain higher-risk but legally supportable sources, we outline here the ethical and legal considerations underlying **MixtureVitae**'s construction. Our goal is not to offer legal advice or definitive interpretations of copyright law, but rather to articulate the principles—fair use, permissive upstream licensing, government-works doctrine, and the EU text-and-data-mining (TDM) (European Union, 2019; Margoni & Kretschmer, 2022) exception—that inform our "permissive-first, risk-mitigated" design philosophy. We provide this discussion so downstream users can understand how specific dataset subsets were evaluated and what residual risks remain despite our filtering and provenance-tracking efforts.

## H.1   Fair Use of Government Works

In order to increase the diversity of our dataset, we included ≈11B tokens of governmental website data from US federal, US non-federal, and non-US government sources. While works created by the US federal government are generally not copyrightable, other governmental website content may neither be expressly in

the public domain nor explicitly licensed. For those sources, we rely on fair use principles (Congress (1976); Lemley & Casey (2017)) and the EU text and data mining exceptions (European Union (2019); Margoni & Kretschmer (2022)), which together mitigate the risk associated with using this subset.

Our ethical and legal reasoning for using this government web content–sourced from Common Crawl–related datasets (Common Crawl Foundation, 2025) that respect *robots.txt* opt-out–is as follows:

- **Public Purpose Alignment**: The content created by governments is normally meant to be shared with the public, and by using the data for training we are assisting this purpose.

- **Purpose of Use**: From a legal perspective, the government works are being redistributed as part of an open source, no-fee dataset, used to create models are less likely to violate copyright. This purpose is clearly not to compete with the government's own usage.

- **Effect on Potential Market**: Our use of government website content is unlikely to affect any potential market for that content, as governments typically do not exploit these materials commercially in ways that would compete with our dataset or downstream models. This factor favors a finding of fair use.

- **Nature of the Content**: The nature of the content is mostly public announcements, content of public interest, governmental functions or the like. Again, we believe there is strong public policy interest for fair use of this type of information.

- **Amount Used**: While we use all or almost all of the content of the government website, the amount of usage is not determinative of fair-use or not fair-use.

- **Federal vs. Non-Federal Works**: Lastly, US works created by the federal governments are generally not copyrightable. However, we recognize that this is not the case for other foreign governmental works, or non-federal works.

For these reasons, we believe using government website data presents relatively lower copyright risk. To further minimize risk–for example, the potential inclusion of third-party copyrighted works embedded in government web pages–we apply keyword filters such as "All Rights Reserved" and "Copyright ©" to exclude pages that contain such terms.

Recent court cases, as of the writing of this paper, include:

- **Bartz v. Anthropic PBC**: district court ruling that use of purchased copies of books for AI training is fair use.

- **Kadrey v. Meta Platforms, Inc.,**: district court ruling that training on authors' books was transformative fair use.

These developments lend some support to the argument that AI training on web-text data—including our relatively small, public-facing government subset—can fall within fair use, though the case law is still evolving.

### H.2 Other Tier-2 Data With Opaque or Mixed Provenance

Similarly, our dataset includes data whose provenance is not entirely transparent even though the license on the upstream dataset appears permissive, such as `The Stack V1` and other Tier-2 sources identified in Appendix K. In the case of `The Stack V1`, a line-by-line audit to remove copyrighted content has not been performed, and therefore some risk remains in its usage. Nonetheless, we rely on fair use to justify this usage because the data are used to train models, rather than to provide a substitutive or competing software product. For a more detailed discussion of `The Stack V1`, see Section I.

For other Tier-2 data, some upstream generator models impose conditions on downstream use—such as the Llama license, which requires model users to adhere to certain limitations. We do not believe we are bound

by terms that were not contractually passed through to us by our direct licensor, although this issue is subject to debate. We therefore classify this small portion ($\approx$4%) of the dataset as **Tier 2(b)**.

There are additional Tier-2 data where the provenance is partially opaque. For example, a small portion of our P3 dataset, when converted into a few-shot format, may pose higher risk than other Tier-1 data. While the ultimate source datasets that constitute P3 are well-known academic benchmarks, some of those component datasets do not provide explicit licenses. Nonetheless, we consider the resulting few-shot datasets to be highly transformative and unlikely to compete with the underlying works: they are mixed and reformatted multiple times for the specific purpose of training classification and few-shot models, rather than, for example, serving as standalone product reviews. We classify these higher-risk works as **Tier 2(b)** and include them in our dataset with that caveat.

### H.3 Reliance on EU Text and Data Mining Exceptions

We also rely, to some extent, on the EU text and data mining exception (European Union, 2019) for our inclusion of web-crawled data. This regime is complementary to US fair-use doctrine, and we mention it here for completeness. In particular, we depend on Common Crawl's practice of respecting *robots.txt* at the time of crawling. We do not believe retroactive recrawling is legally necessary to determine whether a work was subsequently opted out, but we nonetheless commend efforts towards doing so, such as Apertus (Hernández-Cano et al., 2025).

### H.4 Residual Copyright and Trademark Risks

The copyright risks in machine learning are complex. For example, copyrighted materials may appear as limited fair-use quotations in Wikipedia articles [6]. A model trained on such materials in the aggregate could, in principle, generate more substantial and potentially infringing text than the short quotations present in the dataset. Future work should address this risk, including (i) copyright evaluation audits of datasets, and (ii) model-level mitigations that encourage limited direct quotation and discourage reproduction of substantial protected passages.

As with other large, permissively licensed datasets, additional legal risks remain, including trademark risks. For instance, while training on a Wikipedia article about "Spiderman" may be relatively low risk (given its CC-BY-SA license and the educational, summarizing nature of the article), a model that subsequently generates new stories featuring the character name "Spiderman"—even if the plots themselves are not derived from existing human-created stories—may still implicate trademark rights. Addressing those issues thoroughly is beyond the scope of this work and is left for future research.

### H.5 International Jurisdictional Fragmentation

The fair-use and TDM-exception arguments above rely primarily on US legal frameworks (17 U.S.C. § 107 and recent district-court rulings in *Bartz v. Anthropic PBC* and *Kadrey v. Meta Platforms, Inc.*) and on the EU Digital Single Market Directive 2019/790. Copyright regimes, however, differ significantly across jurisdictions. Several jurisdictions lack a general fair-use doctrine entirely or impose narrower conditions on text-and-data-mining activities; the legal status of government works, *robots.txt* compliance, and the downstream obligations attached to generator-model licenses can also vary. We therefore caution that the risk assessment implied by our three-tier scheme is not automatically transferable to all legal environments. International adopters–particularly commercial users in jurisdictions outside the US and EU–should consult local legal counsel before deploying models trained on **MixtureVitae** subsets whose permissibility may depend on jurisdiction-specific doctrines (e.g., government works, web-crawled Tier-2 content, or Tier 2(b) data derived from generator models with restrictive community licenses).

We do not and cannot guarantee that, even with rigorous provenance tracking and standard filtering, the dataset is free of legal risk. Nothing in this section constitutes legal advice. We recommend that anyone who

---

[6]https://en.wikipedia.org/wiki/Wikipedia:Quotations#Copyrighted_material_and_fair_use

uses our datasets consult their own legal counsel in their jurisdiction before deploying models trained on this data in commercial settings.

## I    Provenance and Rationale for The Stack v1 (OpenRAIL-M and terms of use)

Our inclusion of 53.2B tokens sourced from **The Stack v1** (Kocetkov et al., 2023), which we categorize by its governing dataset card terms of use and which subsequent model uses the **OpenRAIL-M** license, warrants this specific note on provenance. The data was included based on the following rationale:

- **Source and Filtering Methodology:** The dataset originates from a large-scale scrape of GitHub. The BigCode project curated this data by applying a filter to include only those repositories that contained a clear permissive license file (e.g., MIT, Apache 2.0, BSD) at the root level.

- **Acknowledged Heuristic:** This repository-level filtering is a *heuristic* and not a file-level guarantee. As acknowledged by the dataset's creators, this process cannot perfectly resolve complex cases of multi-licensing within a single repository, such as the inclusion of non-permissively licensed vendor libraries or mixed-license assets alongside permissively-licensed code.

- **Inclusion Justification:** Despite this caveat, The Stack v1 represents the largest-available public corpus curated with the *explicit goal* of permissive filtering. Excluding it would make training a high-performance, open, and risk-mitigated code model nearly impossible. Its "best-effort" permissive curation philosophy directly aligns with our dataset's core principle of risk-mitigation.

Thus we include it in our dataset with the classification of Tier-2, as defined in Section 2.1.4.

## J    Data Filtering Reasoning and Protocol

To promote transparency, we describe our protocol for defining and checking the lists and content of the pseudo-crawled portion of MixtureVitae.

### J.1    Governmental and NGO Domain Patterns

The following list of URL patterns was used to filter for governmental, non-governmental, and international organization websites from the web datasets. We gathered the list by examining public records, Wikipedia lists, and the like. The list is not as simple as **gov.** because international governments use different TLDs. Moreover, some spam websites masquerades as **.gov** websites. Two of the authors examined each domain either online or through the *Internet Archives' Wayback Machine* to confirm they belonged to a government website. After performing a pseudo-crawl on FineFineWeb, Nemotron-CC and MGACorpus, the authors manually audited the data for quality, and filtered out spam websites with similar website names, which were added to blocklists.

The **.gov, .gov/, and .mil/** websites are US Federal governmental works. To the extent we could, we filtered any sites that had keywords indicating reservations of rights. We believe this lowers the risk of inadvertent third party copyrighted works appearing on US Federal works, and is in the spirit of the EU text data mining opt-out conventions. We also note that the ultimate source of these websites is from Common Crawl which already also respects the *robots.txt* opt-out.

- `gov` (as a suffix)
- `gov/`
- `mil/`

All other websites in this category are specifically international governments or NGOs.

`vlada.mk, vlada.cz, kormany.hu, regeringen.*, rijksoverheid.nl, government.nl, bund.de, bundesregierung.de, government.ru, gc.ca, admin.ch, www.gob.cl/, www.gob.ec/, guatemala.gob.gt/, presidencia.gob.hn/, www.gob.mx/, presidencia.gob.pa/, www.gob.pe/, gob.es/, argentina.gob.ar/, tanzania.go.tz/, indonesia.go.id/, go.kr/, go.jp/, thailand.go.th/, europa.eu/, un/, int/, govt., www.gub.uy, gov., gouv.`

### J.2 Curated Permissive Domain List

The following list of approximately 50 domains was curated based on their known public domain or CC-BY-SA* license status or a permissive status. The websites were chosen for their diversity of content. Two of the authors–one of which has a legal background–examined the websites' terms of use, or relevant sections online or on the Way Back Machine to confirm licensing and permission status. After performing a pseudo-crawl on FineFineWeb, Nemotron-CC and MGACorpus, the authors manually reviewed the data for quality, and filtered out spam websites with similar website names as the below. These spam sites were added to blocklists.

```
free.law, europeana.eu, publicdomainreview.org, wisdomcommons.org, intratext.com, mediawiki.org, wikimedia.org,
wikidata.org, wikipedia.org, wikisource.org, wikifunctions.org, wikiquote.org, wikinews.org, wikivoyage.org,
wiktionary.org, wikibooks.org, courtlistener.com/[7], case.law, pressbooks.oer.hawaii.edu, huggingface.co/docs,
opencourselibrary.org, medbiq.org, doabooks.org, bccampus.ca, open.umn.edu/opentextbooks, www.gutenberg.org,
mozilla.org, www.eclipse.org, apache.org, python.org, pytorch.org, numpy.org, scipy.org, opencv.org, scikit-learn.org,
pydata.org, matplotlib.org, palletsprojects.com, sqlalchemy.org, pypi.org, sympy.org, nltk.org, scrapy.org, owasp.org,
creativecommons.org, wikia.com, foodista.com, fandom.com, attack.mitre.org
```

The vast majority of these sites are CC-BY licensed. However, there are some that have other open licenses as shown in Table 14.

Table 14: Software Licenses and Associated Websites

| License | Websites |
|---|---|
| BSD 3-Clause | scipy.org, sympy.org, matplotlib.org, scrapy.org, scikit-learn.org, pydata.org, pytorch.org, palletsprojects.com |
| Mozilla Public License | mozilla.org |
| Python Software Foundation License 2.0 | python.org |
| Apache 2.0 | apache.org, nltk.org, opencv.org |
| MIT License | sqlalchemy.org |
| Eclipse Public License | www.eclipse.org |
| MedBiquitous Standards Public License | medbiq.org |

## K  Synthetic Data Source Provenance

To ensure full transparency regarding the "permissive-first" nature of **MixtureVitae**, we provide a detailed provenance audit of our synthetic data components in Table 15, including classification to tiers as defined in Section 2.1.4.

Table 15: Detailed provenance of synthetic data sources in **MixtureVitae**.

| Dataset Name | Dataset License | Model | Seed Data Provenance | Token Count(B) | Notes |
|---|---|---|---|---|---|
| **Tier 1: Fully Permissive (≈ 161B Tokens)** | | | | | |
| GlaiveAI Reasoning | Apache 2.0 | Permissive | N/A | 38.366 | Fully synthetic |
| Nemotron (Science & Math) | CC-BY-4.0 | Permissive | Permissive (StackOverflow, WildChat) | 22.310 | Science & Math subset of Llama-Nemotron-Post-Training-Dataset |
| Ling-Coder/SyntheticQA | Apache 2.0 | Permissive | N/A | 19.852 | |
| Open Thoughts | Apache 2.0 | Permissive | Permissive (OpenMath-2-Math, CodeGolf, OpenCode, etc) | 18.786 | Excludes Organic Chemistry subset |
| EuroPat | Public Domain | Permissive | Permissive | 11.586 | Synthetic image captions created from patents |
| P3 (Permissive Subset) | Apache 2.0 | N/A | Permissive (ARC, PIQA, BoolQ, etc) | 10.130 | |
| Nemotron-CC | Common Crawl ToS | Permissive | Permissive (Common Crawl) | 6.230 | Using a Permissive-only subset |

*Continued on next page*

---

[7]For `courtlistener.com`, the terms of use says it is CC-BY-ND, but the underlying court cases are public domain, and the content from this website is merely 176KB and is de minimis.

*Table 15 continued from previous page*

| Dataset Name | Dataset License | Model | Seed Data Provenance | Token Count(B) | Notes |
|---|---|---|---|---|---|
| YouTube | CC-BY-4.0 | Permissive | Permissive (VALID, CommonCorpus) | 7.386 | Derived from CC-BY licensed YouTube content |
| Prism-Math | CC-BY-4.0 | Permissive | Permissive (NuminaMath-1.5) | 5.682 | |
| DeepMind Math | Apache 2.0 | N/A | Permissive (Procedurally Generated) | 4.232 | |
| Misc. Instruct. / NVidia OpenMathInstruct-1 | NVIDIA license | Permissive | Permissive (GSM8K, MATH) | 2.440 | |
| Websights | CC-BY-4.0 | Permissive | N/A | 1.018 | Fully synthetic |
| Misc Instruct. / MetaMathQA-R1 (responses) | MIT | Permissive | Permissive (GSM8K, MATH) | 0.672 | |
| Math Word Problems | Apache 2.0 | N/A | Permissive | 0.456 | Procedurally Generated |
| Ling-Coder/DPO | Apache 2.0 | Permissive | Unknown (Common-Crawl) | 0.398 | |
| Misc. Instruct. / OpenR1-Math-220k | Apache 2.0 | Permissive | Permissive (NuminaMath-1.5) | 0.320 | |
| Misc. Instruct. / NVIDIA SFT Datablend | CC-BY-4.0 | Permissive | Permissive (MNLI, COPA, PIQA, etc) | 0.286 | |
| Misc. Instruct. / OpenThoughts-114k-Code (decontaminated) | Apache 2.0 | Permissive | Permissive (TACO, Apps , CodeContests, etc) | 0.150 | |
| Misc. Instruct. / Synthetic Code Generations | Apache 2.0 | Permissive | N/A | 0.104 | Fully synthetic |
| Misc. Instruct. / PrimeIntellect StackExchange QnA | Apache 2.0 | N/A | N/A | 0.076 | |
| Misc. Instruct. / PrimeIntellect Real World SWE Problems | Apache 2.0 | Permissive | Permissive (CommitPack) | 0.006 | |
| Misc. Instruct. / PrimeIntellect Synthetic Code Understanding | Apache 2.0 | Permissive | N/A | 0.004 | Fully synthetic |
| Misc. Instruct. / GSM8K (train) | MIT | Permissive | N/A | 0.004 | Fully synthetic |
| **Tier 2(a): Permissive with Upstream Opacity (≈ 35B Tokens)** | | | | | |
| OS-Q2 (OpenScience) | CC-BY-4.0 | Permissive | N/A | 17.366 | Fully synthetic |
| Ring-lite SFT Data | Apache 2.0 | Permissive | Permissive (CodeContest, APPS, TACO, etc) | 14.968 | |
| PyEdu Reasoning | Stack V1, ODC-BY | Permissive | Permissive (The Stack V1) | 3.138 | |
| Misc. Instruct. / Magpie-Phi3-Pro-1M-v0.1 | N/A | Permissive | N/A | 0.386 | Fully synthetic |
| MegaWika | CC-By-SA/4.0 | Permissive | Permissive (Wikipedia) | 0.356 | |
| Misc. Instruct. / Magpie-Qwen2.5-Coder-Pro-300K-v0.1 | N/A | Permissive | N/A | 0.120 | Fully synthetic |
| Misc. Instruct. / NovaSky-AI Sky-T1 | Apache 2.0 | Permissive | Permissive (AIME, MATH, etc) | 0.080 | |
| Misc. Instruct. / BigCode Self-OSS-Instruct | Stack v1 | Permissive | Permissive | 0.018 | |
| Misc. Instruct. / UltraFeedback | MIT | Permissive | Permissive (UltraChat, TruthfulQA, etc) | 0.014 | |
| Misc. Instruct. / CaseHOLD (Phi4 Reasoning Traces) | N/A | Permissive | Permissive (CaseHOLD) | 0.002 | Case law is public domain |
| **Tier 2(b): Restricted, Mixed or Opaque Provenance (≈ 17B Tokens)** | | | | | |
| Ling-Coder/SFT | Apache 2.0 | Permissive | Partially Unknown (Github, CommonCrawl, The Stack, etc) | 7.668 | Unknown provenance of CommonCrawl subset |
| P3 (Commercial Subset) | Apache 2.0 | N/A | Ambiguous-license (Amazon, Rotten Tomatoes, IMDb, etc) | 5.730 | User-generated content on commercial platforms with no verifiable licenses |
| Misc. Instruct. / OpenMathInstruct-2 | CC-By-4.0 | Restricted | Permissive (GSM8K and MATH) | 3.202 | Llama-generated responses |

*Table 15 continued from previous page*

| Dataset Name | Dataset License | Model | Seed Data Provenance | Token Count(B) | Notes |
|---|---|---|---|---|---|
| Misc. Instruct. / OpenManus-RL | Apache 2.0 | Mixed | Permissive (AgentTraj-L, Agent-FLAN, etc) | 0.024 | GPT-4 used for generating traces in the AgentTraj-L subset |
| **Tier 3: Civic / Governmental Works (≈ 15B Tokens)** | | | | | |
| Nemotron-CC (Gov. Portion) | Common Crawl ToS | Permissive | Permissive (Common Crawl) | 8.531 | Using a Permissive-only subset |
| MGACorpus | ODC-By | Permissive | Permissive (FinewebEdu-dedup) | 6.596 | Using only permissive subsets |

To validate the robustness of our permissive-first strategy, we further analyze the contribution of synthetic components categorized as **Tier 2(b)**. This small part of **MixtureVitae** is comprised of subsets which are permissively licensed (e.g., Apache 2.0) but are derived from generator models with restrictive community licenses (such as Llama-3) or seed data with partially opaque origins. As illustrated in Figure 13, removing these Tier 2(b) components yields a training trajectory indistinguishable from the full **MixtureVitae** baseline. This result confirms that our model's strong performance is driven by its core, fully verifiable permissive sources, ensuring that users with strict compliance requirements can safely exclude Tier 2(b) data without compromising downstream quality.

See Section H for a further discussion on our justification for including Tier 2 and in particular Tier 2(b) data.

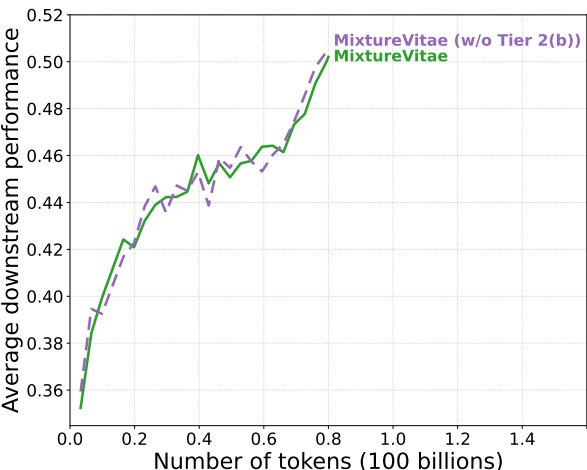

Figure 13: **Ablation of Tier 2(b) components.** We compare the training trajectory of the full **MixtureVitae** dataset (green) against a version excluding Tier 2(b) (purple dashed). Tier 2(b) consists of synthetic data derived from non-permissive generators (e.g., Llama-3) or seeds with opaque provenance. The nearly identical performance curves demonstrate that users requiring strict permissive compliance can exclude these components with negligible impact on downstream model quality.

# L  Scaling Outlook and Future Directions

While **MixtureVitae** currently comprises **422** billion tokens–a scale smaller than frontier runs which often exceed 10 trillion tokens–our primary objective in this work was to establish a proof-of-concept for data efficiency and strong downstream performance within a strict permissive-first, risk-mitigated licensing framework. We identify several concrete avenues to scale this approach to the multi-trillion token regime required for larger foundation models:

**Subset Upsampling.** Standard industry recipes for large-scale training often heavily upsample high-quality data. For instance, Llama 3 (Meta, 2024) employs upsampling factors of 4–10× for its highest-quality subsets

to reach its training budget. In contrast, the current iteration of MixtureVitae does not assign aggressive upsampling factors to individual shards. Applying standard upsampling techniques to our highest-value subsets (such as curated reasoning) would immediately scale their contribution to the total token count.

**Multilingual Expansion.** The current release of MixtureVitae is primarily English-centric. Expanding the sourcing strategy to include multilingual data represents an order-of-magnitude opportunity for scaling. This can be achieved through two primary methods: (1) identifying and allowing international permissively licensed sources, and (2) using machine translation to expand the existing data in MixtureVitae.

**Synthetic Expansion.** Our Math and Reasoning synthetic subsets are generated procedurally or via LLMs. This generation process is horizontally scalable. By increasing the compute budget for generation, these high-density subsets can be expanded significantly without incurring the legal risks associated with scraping organic web data.

**Web Data Rephrasing.** Recent work has demonstrated the utility of rephrasing web data to improve quality and standardize style (Maini et al., 2024). Applying a similar rephrasing pipeline on top of the MixtureVitae web data processing pipeline can further expand the corpus volume while maintaining the strict safety and licensing standards defined in our framework.

## M  Author contributions

- **Huu Nguyen**: Overall lead, data pipeline engineering, data curation, dataset composition, data review, decontamination analysis, policy design, paper co-writing

- **Victor May**: Paper writing lead, project coordination, policy design, decontamination protocol design and analysis

- **Harsh Raj**: Led model training, evaluations and ablation experiments, decontamination analysis, paper co-writing

- **Marianna Nezhurina**: Model training, established major parts of the dataset tokenization and training infrastructure (Megatron-LM container based workflow), conducted scaling tests for distributed training, wrote routines for evaluation based on **lm-eval-harness**, conducted model training, performed evaluations, ablations and paper co-writing

- **Yishan Wang**: Paper co-writing

- **Yanqi Luo**: Paper co-writing

- **Minh Chien Vu**: Dataset curation, multimodal text generation

- **Taishi Nakamura**: Implemented checkpoint conversion routines from Megatron to HuggingFace format, paper co-writing

- **Ken Tsui**: Data curation, filtering and classifier training

- **Van Khue Nguyen**: Dataset curation, multimodal text generation

- **David Salinas**: Performing evaluations and visualizations, paper co-writing

- **Aleksandra Krasnodębska**: Red teaming evaluations, paper co-writing

- **Christoph Schuhmann**: Synthetic math word problems dataset co-design, advising

- **Mats Leon Richter**: Dataset curation and processing, advising

- **Xuan-Son (Sonny) Vu**: Advising and paper co-writing

- **Jenia Jitsev**: Supervision, compute resource acquisition. Reference baseline experiments (open-sci-ref) design and model training. Organized dataset acquisition and transfer across the supercomputers. Co-established environments for experiments across various supercomputers. Experiments and evaluations co-design, results analysis. Paper writing co-lead.

Table 16: Author contributions to this work. Large squares (■) indicate a major contribution and plus symbols (+) indicate a supporting contribution.

| | Dataset Curation & Processing | Policy | Decontamination | Model Training | Evaluation & Red Teaming | Writing | Visualization | Leadership & Coordination | Advising |
|---|---|---|---|---|---|---|---|---|---|
| Huu Nguyen | ■ | ■ | ■ | | | + | | ■ | |
| Victor May | | + | ■ | | | ■ | ■ | ■ | |
| Harsh Raj | | | + | ■ | ■ | + | + | | |
| Marianna Nezhurina | | | + | ■ | ■ | + | + | | |
| Yishan Wang | | | | | | ■ | | | |
| Yanqi Luo | | | | | | + | ■ | | |
| Minh Chien Vu | + | | | | | | | | |
| Taishi Nakamura | | | | + | | + | | | |
| Ken Tsui | + | | | | | | | | |
| Van Khue Nguyen | + | | | | | | | | |
| David Salinas | | | | | + | ■ | + | | |
| Aleksandra Krasnodębska | | | | | + | | | | |
| Christoph Schuhmann | + | | | | | | | | + |
| Mats Leon Richter | + | | | | | | | | + |
| Xuan-Son (Sonny) Vu | | | | | | + | | | + |
| Jenia Jitsev | + | | | ■ | ■ | ■ | + | ■ | ■ |

