# OpenReview forum: "MixtureVitae: Open Web-Scale Pretraining Dataset With High Quality Instruction and Reasoning Data Built from Permissive-First Text Sources"
_TMLR — Accepted by TMLR_

### Review · Reviewer_i6U4 · 2026-02-23

**Summary Of Contributions:**

The paper introduces MixtureVitae, a 422-billion-token open-access pretraining dataset designed to minimize legal and copyright risks while achieving competitive downstream performance. The authors present a "permissive-first" and risk-mitigated curation strategy that categorizes data into three auditable tiers: explicit open licenses/public domain, curated permissive corpora with some upstream opacity, and civic/governmental works.
A core contribution is demonstrating that heavily front-loading high-quality instruction and reasoning data—largely synthetic and generated from permissive sources—can compensate for the absence of massive, indiscriminately scraped web data.

Key Strengths:
- Strong Empirical Results: The 1.7B parameter base model trained on 300B MixtureVitae tokens matched or exceeded a 1.7B instruction-tuned baseline (SmolLM2) trained on 11T tokens across key math and coding benchmarks (GSM8K, HumanEval, MBPP).
- Rigorous Evaluation: The authors utilize the controlled open-sci-ref training protocol (fixed architectures and hyperparameters across 130M to 1.7B parameter scales) to effectively isolate the dataset's impact.
- Thorough Decontamination: The paper includes a comprehensive 13-gram decontamination analysis to prove that the strong math and code performance is not an artifact of test-set leakage.

Key Weaknesses:
- Scale Limitations: The total dataset size (422B tokens) is relatively small compared to modern frontier models, which often train on trillions of tokens, though the authors do provide a scaling outlook.
- Residual Legal Risk: The paper relies heavily on fair-use arguments for certain data inclusions (e.g., government works and heuristically filtered code repositories), meaning the dataset is "risk-mitigated" rather than strictly "risk-free".
- Generator Model Licensing: A small portion of the dataset (~4%) relies on synthetic data generated from models with more restrictive licenses (like Llama-3), though ablation studies show removing this data does not harm performance.

**Audience:**

Yes

**Audience Explanation:**

The machine learning community is facing increasing legal and ethical scrutiny regarding the mass scraping of copyrighted web data for foundation models. Practitioners and researchers are actively seeking viable alternatives. This paper is highly relevant because it provides concrete evidence that reliance on legally ambiguous data is not a strict prerequisite for training capable LLMs. Additionally, the insight that front-loading reasoning data during pretraining can radically improve token efficiency—achieving results at 300B tokens that normally require trillions—is highly valuable to researchers operating with constrained compute budgets.

**Broader Impact Concerns:**

The authors include a Broader Impact Statement (Appendix B) and a discussion on residual copyright and trademark risks (Appendix H.4). They also perform a red-teaming safety evaluation (ToxiGen, Do-Not-Answer, AdvBench), showing their model performs competitively with baselines.

However, the paper heavily promotes "front-loading" reasoning and instruction capabilities into the base model. While this boosts math and coding performance, models with advanced reasoning capabilities built directly into the base layer can sometimes be easier to inadvertently steer toward harmful logic or malicious code generation before alignment (RLHF/DPO) takes place. A brief addition to the Broader Impact Statement acknowledging the dual-use risks of front-loading reasoning in base models would sufficiently address this.

**Claims And Evidence:**

Yes

**Claims Explanation:**

The authors systematically support their claims through a highly controlled experimental framework. By using the open-sci-ref procedure, they freeze model architectures and training hyperparameters, ensuring that any performance variations are strictly attributable to the dataset composition. They benchmark MixtureVitae against a robust suite of both permissive and non-permissive/mixed-license datasets.

**Requested Changes:**

- Nuance on Fair Use: The authors clearly outline their position on using governmental works under fair use. However, international copyright law is highly fragmented. It would strengthen the paper to explicitly state that the fair-use arguments presented primarily lean on US legal paradigms, and international adopters should exercise caution based on local jurisdictions.

---

> ### Author Response · Authors · 2026-03-20
> **Rebuttal Response Reviewer i6U4**
>
> We thank the reviewer for the thorough and positive assessment, and for highlighting our strong empirical results, rigorous evaluation methodology, and comprehensive decontamination analysis. We appreciate the detailed summary of our contributions and address each relevant below.
>
> ### Remark 1: Scale limitations (422B tokens)
>
> We acknowledge the observation regarding dataset size relative to frontier training runs. As discussed in our responses to Reviewers x5Vt and gtWs, the 422B unique token count places MixtureVitae as the largest permissive pretraining corpus in terms of unique content — larger than Comma-0.1 (~316B unique tokens, despite training on ~1T effective tokens due to up to 6× repetition).
>
> It is a standard practice to validate a new dataset mixture at smaller scales first. This is the approach taken by The Common Pile / Comma-0.1 (concurrent work) and by influential earlier work (DCLM). Our primary goal is a proof-of-concept demonstrating that a permissive-first, reasoning-heavy mixture can achieve performance that is competitive with mixed-license baselines — and the results confirm this strongly. We provide a detailed scaling outlook in Appendix K, outlining concrete paths (upsampling, multilingual expansion, synthetic growth, web rephrasing) to extend MixtureVitae to the multi-trillion-token regime.
>
> ### Remark 2: Residual legal risk — "risk-mitigated" rather than "risk-free"
>
> The reviewer correctly identifies our intentional design choice. We deliberately use the term "permissive-first, risk-mitigated" rather than "risk-free" throughout the paper. The tiered provenance framework (Tiers 1, 2, 2b) is specifically designed to give users flexibility: researchers with different institutional contexts, legal requirements and risk tolerances can select which tiers to include. For example, a research lab in the U.S. may be comfortable including all tiers, while a commercial user in a more restrictive copyright regime may restrict to Tier 1 only. Our ablation studies (Section 3.4, Figure 5) show that the performance impact of excluding higher-risk tiers is modest, supporting such selective use.
>
> ### Remark 3: Generator model licensing (~4% Tier 2b data)
>
> We appreciate the reviewer noting that our ablation studies demonstrate removing this subset does not harm performance. This is a key finding: users who wish to avoid any data generated by more restrictively licensed models (such as Llama-3) can exclude Tier 2(b) without sacrificing capability. The explicit tier labeling makes this straightforward.
>
> ### Requested Change: Nuance on fair use and international law
>
> We agree this is an important clarification. While our Appendix H already discusses both US fair use doctrine and the EU text-and-data-mining exception (Directive 2019/790), the reviewer is right that we should more prominently acknowledge the international fragmentation of copyright law. We will add an explicit statement in both the main text and Appendix H noting that:
>
> 1. Our fair-use arguments primarily rely on US legal frameworks and the EU TDM exception.
> 2. Copyright regimes differ significantly across jurisdictions — for instance, some jurisdictions lack a general fair-use doctrine or impose different conditions on text-and-data-mining activities.
> 3. International adopters should consult local legal counsel before deploying models trained on datasets containing government works or other content whose permissibility may vary by jurisdiction.
>
> We note that this caveat is already partially present in the final paragraph of Appendix H ("We do not and cannot guarantee that [...] the dataset is free of legal risk. [...] We recommend that anyone who uses our datasets consult their own legal counsel in their jurisdiction"), but we will make it more prominent and specific to the international dimension.
>
> ### Broader Impact: Dual-use risks of front-loading reasoning in base models
>
> We appreciate this thoughtful observation. We note that base models are by definition not aligned — alignment is the task of post-training (RLHF, DPO, etc.) and is not the focus of this study. All base models, regardless of their pretraining mixture, carry dual-use risks prior to alignment.
>
> The reviewer raises the specific question of whether front-loading reasoning capabilities into the base model makes alignment more difficult or creates additional risk. This is an important and currently open research question. It is unclear whether reasoning-capable base models are harder to align, or whether, conversely, their stronger instruction-following and reasoning foundations actually facilitate more effective alignment. Addressing this question rigorously would require dedicated alignment experiments that are beyond the scope of this dataset-focused work.

---

> > ### Author Response · Authors · 2026-03-20
> > **Rebuttal Response Reviewer i6U4 Continuation**
> >
> > We will add a brief discussion to the Broader Impact Statement acknowledging: (i) that all base models carry dual-use risks prior to alignment, (ii) that the effect of front-loaded reasoning capabilities on alignment difficulty is an open question worthy of further investigation, and (iii) that MixtureVitae's tiered structure allows users to control the extent of reasoning data inclusion based on their safety considerations.

---

### Review · Reviewer_gtWs · 2026-03-06

**Summary Of Contributions:**

**Summary**

The paper introduces MixtureVitae, a new pretraining corpus that uses only permissible data in order to minimize legal risk. The paper divides corpus permissibility into multiple tiers, ranging from low-risk, which contains permissibly licensed text, to higher-risk text, which contains synthetic instructions and reasoning data, along with text whose provenance is not well-defined. The paper trains multiple pretrained models with various token budgets and shows that the performance of these pretrained models is on par with models trained on corpora with non-permissible licenses.

**Strengths**

- The paper is well written. The motivation behind MixtureVitae is clear. The related work is also well organized and covers most of the relevant work.
- Data curation pipeline: The paper does an excellent job curating datasets and identifying their provenance. Table 14 clearly lists the licenses and tiers for all datasets. This work will be very useful for practitioners.
- Experiments: The pretrained models trained on MixtureVitae show stronger performance than models trained on other permissible datasets.

**Weaknesses**

- The paper needs to justify why this work differs from Kandpal et al. (2025), as they share the same motivation. However, they introduce a dataset with over a trillion tokens. In contrast, the proposed work has a corpus with 400B tokens. Elaborating on the key differences, both in terms of dataset mixture and performance, would be helpful to readers. Furthermore, are there reasons why the proposed pretraining corpus cannot be scaled to a trillion tokens? Suppose we wish to train a Chinchilla-optimal model. The largest model one could train on the MixtureVitae corpus is an LLM with 21B parameters. This is a potential limitation.

- Figure 5 shows that instruction-tuning data plays a major role in achieving strong performance. This suggests that the web data is not the primary reason for the strong performance in Table 2. Wouldn’t this be an unfair comparison to other permissible pretraining corpora that do not include instruction or reasoning data?

- Reporting performance: While this is not strictly a limitation, I noticed a few discrepancies in the results reported in the paper. First, why does MixtureVitae (green) have different performance in Figure 3a and Figure 4a? The numbers reported in Figure 3a indicate the model’s performance is around 0.55, whereas Figure 4a shows around 50. Furthermore, I found it a little strange that Figure 5 evaluates on 10 evaluation tasks rather than 11 target tasks (Figures 3a and 4a). This could be a typo. I just wanted to double-check.

**Audience:**

Yes

**Audience Explanation:**

The paper creates a new pretraining corpus by collecting datasets with permissive licenses. This work would be useful for the TMLR audience interested in pretraining.

**Claims And Evidence:**

Yes

**Claims Explanation:**

Yes, the claims made in the paper are accurate. I have asked for additional clarification in the summary of contributions section.

**Requested Changes:**

In addition to addressing the weaknesses, which are far more critical, it would be great if you incorporated these changes to strengthen the paper:

- It would improve readability if Figures 1(a) and (b) used two distinct color palettes. Since the colors in both charts are the same (or similar), one could misinterpret them when skimming.

- It would be great if there were a scatter plot of all models trained under different permissible licenses, with the x-axis as the number of tokens and the y-axis as average performance.

- Table 14 in the appendix contains overlapping text.

---

> ### Author Response · Authors · 2026-03-20
>
> We thank the reviewer for the constructive assessment, and for highlighting the quality of our writing, curation pipeline, and experimental results. We are glad the reviewer found Table 14 and our provenance documentation useful for practitioners. We address relevant points below.
>
> ### Remark 1: Justifying the difference from Kandpal et al. (2025) / The Common Pile and scalability
>
> We appreciate the request to clarify the relationship to The Common Pile (Kandpal et al., 2025). As detailed in our response to Reviewer x5Vt, the training dataset derived from The Common Pile — Comma-0.1 — is one of our two central permissive baselines (Section 3.1), and the comparison features prominently throughout the paper (Table 1, Table 2, Figure 3, Appendix Figures 7–9).
>
> A key point regarding dataset scale: while Comma-0.1 trains on \~1T tokens, this is achieved through substantial repetition (up to 6× for most sources, as documented in Table 7 of Kandpal et al., 2025). The unique token count in Comma-0.1 is approximately **~316B tokens**. MixtureVitae contains **422B unique tokens** — ~33% more unique data. Thus, MixtureVitae is in fact the larger permissive corpus in terms of unique content. Other permissive datasets in our comparison (CommonCorpus) also have comparable or smaller unique token budgets.
>
> Regarding scalability: it is standard practice to first validate a dataset mixture's potential at smaller scales before scaling up. This approach is taken by Comma-0.1 itself and by earlier influential work such as Phi-4 and DCLM. We outline concrete scaling paths in Appendix K: subset upsampling (MixtureVitae currently does not apply special upsampling factors, unlike e.g. Llama 3 which upsamples high-quality subsets 4–10×), multilingual expansion, synthetic data growth, and web data rephrasing. These paths can readily extend MixtureVitae to the multi-trillion-token regime, enabling Chinchilla-optimal training of substantially larger models. We will expand this discussion in the revised manuscript.
>
> ### Remark 2: Fairness of comparison — role of reasoning and instruction data
>
> This is an important point. We wish to clarify two aspects:
>
> First, our comparison is not limited to permissive pretraining corpora. We also compare against strong **non-permissive** datasets that themselves incorporate reasoning and synthetic data in their mix — notably Nemotron-CC-HQ (Su et al., 2025), which includes substantial amounts of synthetically rewritten and curated data. As shown in Table 2, MixtureVitae achieves a dramatic advantage over Nemotron-CC-HQ on math/code tasks (e.g., GSM-8K: 0.53 vs 0.03; MBPP: 0.38 vs 0.00 at 300B tokens). This advantage persists even when comparing against Nemotron-CC-HQ trained on 1T tokens (GSM-8K: 0.03, MBPP: 0.04). It is a striking and surprising finding that datasets like FineWeb-Edu and Nemotron-CC, despite having substantial math/code data, lag so far behind MixtureVitae on reasoning benchmarks.
>
> Second, even permissive datasets like Comma-0.1 contain fractions of curated instruction-like data. The Common Pile includes code (Stack V2, GitHub Archive), question-answering data (Data Provenance Initiative, StackExchange), and curated academic text — sources that serve similar functional roles to instruction data. Therefore, the comparison cannot be considered unfair, as these types of subsets are present across baselines, and MixtureVitae strongly outperforms them all.
>
> Our methodology is designed to answer the question: *how far can one go by changing only the pretraining corpus, holding architecture and training budget fixed, under a permissive-first framework?* The inclusion of reasoning data in pretraining follows validated approaches like Phi-4 (Abdin et al., 2024). We additionally compare to the post-trained SmolLM2-1.7B-Instruct (trained on ~11T tokens), which our 300B-token base model matches or exceeds on GSM-8K, HumanEval, and MBPP (Table 2). This comparison is, if anything, unfavorable to MixtureVitae: SmolLM2-1.7B-Instruct is a strong instruction-tuned model that was both pre-trained and post-trained with large amounts of math/code/instruction/reasoning data on ~36× more tokens — yet our single-stage base model still matches or outperforms it.

---

> > ### Author Response · Authors · 2026-03-20
> > **Rebuttal Response Reviewer gtWs Continuation 1**
> >
> > ### Remark 3: Reporting performance — figure discrepancies
> >
> > We thank the reviewer for the careful reading. These are not errors/typos:
> >
> > **Figure 3a vs Figure 4a**: Figure 3a shows average performance for the **300B token** training budget, while Figure 4a shows results at the **100B token** budget (used for the decontamination validation). The different performance levels (~0.55 vs ~0.50) reflect the expected performance difference between these two training scales. Additionally, the y-axis scale conventions differ (0–1 in Figure 3a; 0–100 in Figure 4a). We will make these distinctions more explicit in the revised captions to avoid confusion.
> >
> > **Figure 5 with 10 tasks vs 11 tasks in Figures 3a/4a**: The ablation study (Figure 5) was conducted with the original 10-task evaluation set (COPA, LAMBADA, OpenBookQA, Winogrande, MMLU, PIQA, ARC-Challenge, ARC-Easy, HellaSwag, BoolQ). CommonSense-QA was added to the main evaluation suite subsequently, bringing the total to 11 tasks in Figures 3 and Table 1. The ablation results in Figure 5a therefore average over 10 tasks, while Figures 3a and 4a average over 11. This is noted in the Figure 5 caption ("Average performance on 10 downstream tasks") but we agree this could be made more explicit. We will add a clarifying note in the revised manuscript.
> >
> > ### Requested Changes
> >
> > - **Distinct color palettes for Figures 1(a) and (b)**: Agreed. We will update the figures with clearly distinct color schemes.
> > - **Scatter plot of tokens vs performance**: This is a useful suggestion. We provide a compute scaling plot showing average performance (y-axis) vs. compute in FLOPs (x-axis) for all permissive datasets (MixtureVitae, Comma-0.1, CommonCorpus) across four model scales (0.13B, 0.4B, 1.3B, 1.7B) trained on 300B tokens. The plot (see Fig. http://tiny.cc/d6n0101) shows a clear advantage of MixtureVitae: both a large offset above the other permissive dataset trends and a stronger scaling slope. For the camera-ready version, we will extend this plot to include mixed-license datasets (e.g., Nemotron-CC-HQ) for a more comprehensive overview.
> > - **Table 14 overlapping text**: Agreed. We will fix the formatting issue.

---

### Review · Reviewer_x5Vt · 2026-03-07

**Summary Of Contributions:**

The paper proposes MixtureVitae, a new curated dataset of risk-stratified data for training LLMs. Data is collected from sources with clear provenance (or provenance is tagged with risk levels to disentangle during pretraining). Several models are trained on this mixture and evaluated on a set of standard (if somewhat old) benchmark datasets. The paper specifically looks at how including more synthetic reasoning/rewrite data impacts performance. They paper also makes efforts to really rule out test set contamination from the benchmarks.

This is a contribution beyond open training datasets like The Common Pile, since it provides structure around training data created with different provenance (e.g., synthetic rewrite data). The paper introduces risk tiers for different sources of data while also adding more instruction and reasoning data to the training mixtures. So, it is not strictly open in the same sense as The Common Pile, but it is a useful test bed for ablating data with different IP considerations.

I like this paper, and I think it speaks to a critical need in LLM research.

**Audience:**

Yes

**Audience Explanation:**

I believe at least part of the TMLR audience would be interested in this paper because it addresses an important and underdeveloped problem: how to construct useful LLM training datasets under meaningful IP and provenance constraints. This is especially relevant given growing interest in legally safer and more auditable training pipelines

**Broader Impact Concerns:**

The paper addresses an important issue around data provenance and legal risk, which is itself a positive contribution. However, I think the broader impact discussion could more clearly acknowledge the tradeoff introduced by including synthetic and rewritten data in the training mixture.

**Claims And Evidence:**

Yes

**Claims Explanation:**

## Strengths

- Datasets that directly take on IP constraints and the challenge of teaching LLMs useful skills without entirely relying on proprietary sources are a fundamental need in LLM research. The paper makes a nice contribution in the form of MixtureVitae.
- I especially like that the paper does not simply present another “open” dataset, but instead provides structure around provenance and legal risk. That makes it more useful for researchers who want to study the impact of different data sources and risk profiles.
- The paper also usefully expands beyond strictly open data in the style of The Common Pile by explicitly incorporating instruction and reasoning data, including synthetic or rewritten content, while still preserving provenance information.

## Weaknesses
- The paper needs stronger narrative framing around how it differs from similar inspired efforts like The Common Pile.
- I am somewhat torn on the increased reasoning and instruction data in the mixtures. Instead of covering broad, heterogeneous topic areas, MixtureVitae commits to a stronger prior over desired skills by including synthetic data. That is not necessarily a flaw, but it does make the dataset feel less like a general-purpose open pretraining corpus and more like a capability-shaped mixture (perhaps no more so than what happens when "raw" sources are picked for inclusion in these types of datasets.
- That said, because the data is provenance-tagged, researchers can still disentangle these choices and more carefully study where skills develop across pretraining, mid-training, and post-training.
- The set of evaluation tasks feels a bit dated and is mostly multiple choice.
- There are no domain-specific evaluations (e.g., law, healthcare).

**Requested Changes:**

- Better articulate the differences from other similar efforts, especially The Common Pile. What is the additive contribution beyond The Common Pile?
- I would include more recent and more challenging evaluation tasks, such as MMLU-Pro, GPQA, LiveBench, etc., along with some domain-specific evaluations if possible (LegalBench, MedHELM).
- More clearly explain the intended identity of the dataset: is it meant to be a general-purpose open pretraining corpus, or a deliberately capability-shaped mixture designed to recover reasoning and instruction-following performance under stricter data constraints?

---

> ### Author Response · Authors · 2026-03-20
> **Rebuttal Response Reviewer x5Vt**
>
> We thank the reviewer for the positive and thoughtful assessment, and for recognizing that MixtureVitae "speaks to a critical need in LLM research" by providing "structure around provenance and legal risk." We are encouraged by the reviewer's appreciation of our contribution beyond simply presenting another open dataset. We address relevant points below.
>
> ### Remark 1: Stronger narrative framing around how MixtureVitae differs from The Common Pile
>
> We agree that clearly articulating the differences from The Common Pile is important, and we appreciate the opportunity to elaborate.
>
> First, we wish to clarify the relationship: The Common Pile (Kandpal et al., 2025) is an 8TB *data pool* of permissively licensed sources. Comma-0.1 is the *training dataset* derived from it, and this is precisely the baseline we compare against extensively throughout the paper. Comma-0.1 is one of our two central permissive baselines (alongside CommonCorpus), as stated in Section 3.1 (Experimental Setup, p. 7). The comparison runs through all our central results: Table 1 (language understanding), Table 2 (reasoning/math/code), Figure 3, and Appendix Figures 7–9.
>
> A key quantitative distinction concerns the effective dataset size. While Comma-0.1 trains on \~1T tokens, this is achieved through substantial repetition of the underlying data: as documented in the Comma-0.1 data mixture (Table 7 of Kandpal et al., 2025), most sources are repeated up to 6×, and the unique token count in Comma-0.1 is approximately **~316B tokens**. MixtureVitae, by contrast, contains **422B unique tokens** — making it a larger corpus in terms of unique content (~33% more unique data).
>
> Beyond scale, MixtureVitae and The Common Pile / Comma-0.1 differ in *design philosophy* and resulting performance of trained models:
>
> - **Data composition**: Comma-0.1 consists primarily of traditional permissive text sources (scientific papers, code, books, government documents). MixtureVitae augments similar permissive sources with a substantial fraction of reasoning and instruction data — following the insight from Phi-4 (Abdin et al., 2024) that front-loading such data into pretraining can dramatically improve capabilities.
> - **Curation strategy**: The Common Pile operates as a broad collection of strictly open sources while avoiding synthetic data due to licensing concerns; MixtureVitae targets a large proportion of synthetic data while applying a tiered provenance framework that categorizes every source by legal risk level, enabling users to selectively include or exclude tiers based on their risk tolerance.
> - **Performance**: As shown in Table 1, MixtureVitae outperforms Comma-0.1 on most language understanding evaluations. The gap becomes dramatic on reasoning/math/code benchmarks (Table 2), where MixtureVitae vastly outperforms all permissive baselines including Comma-0.1 and also exceeds strong non-permissive baselines (FineWeb-Edu, DCLM, Nemotron-CC v1).
>
> We will strengthen the narrative framing in the revised manuscript to make these distinctions more explicit.
>
> ### Remark 2: Increased reasoning and instruction data — general-purpose vs capability-shaped mixture
>
> We appreciate this nuanced observation. While MixtureVitae does include an overproportionally large fraction of reasoning, instruction, math, and code data (including synthetic data), one of the central points of our work is to demonstrate that this does *not* come at the cost of general-purpose capabilities.
>
> Specifically, our results on standard language understanding evaluations — the conventional benchmarks used for evaluating base models — show that:
>
> - **Table 1 and Figure 3**: Models trained on MixtureVitae achieve better results on language understanding tasks than other permissive general-purpose corpora (Comma-0.1, CommonCorpus) and come close to leading non-permissive datasets (DCLM, FineWeb-Edu).
> - **Table 2**: Simultaneously, MixtureVitae strongly outperforms all baselines — both permissive and non-permissive — on reasoning/math/code evaluations.
>
> Thus, MixtureVitae is best described as a **permissive general-purpose pretraining dataset with strongly enhanced reasoning/math/code capabilities** — it does not sacrifice breadth for depth. As stated in the Discussion section: *"We thus provide evidence that heavily increasing reasoning and instruction data fraction on expense of generic web text creates overall boost in performance without hurting core language understanding capabilities."*
>
> As the reviewer notes, because the data is provenance-tagged, researchers can disentangle these choices and study how different data tiers contribute to different capabilities — which is an additional advantage of MixtureVitae's design.

---

> > ### Author Response · Authors · 2026-03-20
> > **Rebuttal Response Reviewer x5Vt Continuation 1**
> >
> > ### Remark 3: Evaluation tasks feel dated and are mostly multiple choice
> >
> > Our evaluation suite is a subset of the DCLM core full set (11 tasks out of the full 22 task set), which is the current state-of-the-art standard for evaluating base models trained at these scales. This same evaluation setup is also used in very recent works, including Nemotron-CLIMB (Parmar et al., 2025; see their Tables 1–3) and nanoChat (https://github.com/karpathy/nanochat?tab=readme-ov-file#time-to-gpt-2-leaderboard), confirming its continued relevance.
> >
> > We note that more advanced evaluations such as MMLU-Pro, GPQA, or LiveBench are primarily designed for larger-scale models and/or post-trained models, and base models at the 130M–1.7B scale that we study would not provide meaningful signal on these benchmarks. We also go beyond the standard base model evaluation suite by reporting scores on GSM-8K, HumanEval, MBPP, and IFEval (Table 2) — benchmarks that are typically reserved for post-trained models. The strong performance of our single-stage base model on these tasks (e.g., 53% on GSM-8K, matching or exceeding the post-trained SmolLM2-1.7B-Instruct) is itself a notable finding.
> >
> > **Additional results on harder benchmarks.** To directly address the reviewer's suggestion, we have conducted additional evaluations on three more recent and challenging benchmarks: **MATH500**, **GPQA Diamond**, and **LiveCodeBench** (pass@1). Results for all 1.7B open-sci-ref models and SmolLM2 baselines are shown below:
> >
> > | Training Dataset | License | Tokens | Training | MATH500 | GPQA-D | LCB |
> > |:---|:---|---:|:---|---:|---:|---:|
> > | | | | | | | |
> > | *open-sci-ref 1.7B, 300B tokens* | | | | | | |
> > | MixtureVitae | permissive | 300B | single-stage pretraining | **0.18** | 0.25\* | **0.13** |
> > | Comma-0.1 | permissive | 300B | single-stage pretraining | 0.00 | 0.23\* | 0.00 |
> > | CommonCorpus | permissive | 300B | single-stage pretraining | 0.00 | 0.24\* | 0.00 |
> > | C4 | mixed | 300B | single-stage pretraining | 0.00 | 0.21\* | 0.00 |
> > | FineWeb-Edu | mixed | 300B | single-stage pretraining | 0.00 | 0.21\* | 0.00 |
> > | SlimPajama | mixed | 300B | single-stage pretraining | 0.00 | 0.22\* | 0.00 |
> > | HPLT-2.0 | mixed | 300B | single-stage pretraining | 0.00 | 0.20\* | 0.00 |
> > | | | | | | | |
> > | *Other models* | | | | | | |
> > | SmolLM2-1.7B | mixed | 11T | multi-stage pretraining | 0.00 | 0.22\* | 0.00 |
> > | SmolLM2-1.7B-Instruct | mixed | 11T | multi-stage pretraining + post-training | 0.17 | **0.29** | 0.05 |
> >
> > *GPQA-D = GPQA Diamond. LCB = LiveCodeBench (pass@1, n_repeats=3). \* At or below random chance (0.25 for 4-way MC).*
> >
> > These results confirm our expectation: harder benchmarks designed for larger or post-trained models yield little to no signal for 1.7B base models. All open-sci-ref baselines — both permissive and non-permissive — score 0.00 on MATH500 and LiveCodeBench. GPQA Diamond scores are at or below random chance (0.25) for all models except SmolLM2-1.7B-Instruct. Notably, MixtureVitae trained model is the **only** that produces non-zero signal on MATH500 (0.18) and LiveCodeBench (0.13), matching SmolLM2-1.7B-Instruct on MATH500 and more than doubling it on LiveCodeBench — despite being a single-stage base model on 36× fewer tokens. We note that, in contrast to the evaluations presented in the paper, these additional benchmarks were not explicitly decontaminated, so results should be interpreted with appropriate caution.
> >
> > ### Remark 4: No domain-specific evaluations (e.g., law, healthcare)
> >
> > Domain-specific evaluations such as LegalBench or MedHELM are beyond the scope of the current work. Our focus is on demonstrating that permissive-first data can be competitive for installing *general-purpose* capabilities during pretraining. Evaluating domain-specific performance would require dedicated domain-specific data curation and possibly larger model scales — both of which are promising directions for future work that we would be happy to acknowledge in the manuscript.
> >
> > ### Requested Change 1: Better articulate differences from The Common Pile
> >
> > We will incorporate the detailed comparison outlined above (unique token count, design philosophy, performance differences) into the revised manuscript to clearly articulate MixtureVitae's additive contribution beyond The Common Pile and Comma-0.1.

---

> ### Author Response · Authors · 2026-03-20
> **Rebuttal Response Reviewer x5Vt Continuation 2**
>
> ### Requested Change 2: More recent and challenging evaluation tasks
>
> As discussed in Remark 3 above, we have now conducted additional evaluations on three more recent and challenging benchmarks — MATH500, GPQA Diamond, and LiveCodeBench — which are exactly the kind of harder tasks the reviewer suggested. The results (see table in Remark 3) confirm that these benchmarks are largely too hard for 1.7B base models, but MixtureVitae is the only model to produce non-zero signal, matching or exceeding SmolLM2-1.7B-Instruct on MATH500 and LiveCodeBench. We will include these results in the revised manuscript alongside the existing DCLM core evaluation suite and the reasoning/math/code benchmarks (GSM-8K, HumanEval, MBPP, IFEval) already reported in Table 2.
>
> ### Requested Change 3: Clarify the intended identity of the dataset
>
> We will expand our framing to explicitly state that MixtureVitae is a **permissive general-purpose pretraining corpus** whose design deliberately front-loads reasoning and instruction data, resulting in models that maintain strong language understanding performance while achieving dramatically enhanced reasoning, math, and code capabilities — even without post-training. This positions MixtureVitae as a general-purpose dataset that also serves as a reasoning-capable foundation, rather than a narrow capability-specific mixture.
>
> ### Broader Impact
>
> We agree that the broader impact discussion should more clearly acknowledge the role of the inclusion of synthetic and rewritten data. The broad communicy consensus is currently that mixing synth and real data creates strong capability boost, with strong evidence for this in various SOTA works, Phi-4 (https://arxiv.org/abs/2412.08905), Rephrasing the web (https://arxiv.org/abs/2401.16380), Nemotron-CC (https://arxiv.org/abs/2412.02595), ReWire (https://arxiv.org/abs/2506.04689) and most recently, FinePhrase (https://huggingface.co/spaces/HuggingFaceFW/finephrase) We will thus add a brief discussion on: (i) the importance of mixing human-authored and synthetic data; (ii) checking the quality of synthetic data generation and similarities and differences compared to quality validation of real text, and (iii) the importance of the provenance-tagged tier structure in enabling users to audit and select both real and synthetic components as needed. We note that MixtureVitae's tiered design is specifically intended to give users transparency and control over these decisions.

---

### Decision · Action_Editor_9D9A · 2026-04-13

**Recommendation:** Accept as is

**Additional Comments:**

All three reviewers recommended acceptance, with one leaning. I think the scope of this paper is beyond that of a typical TMLR submission in terms of the effort involved, representing an especially significant contribution. It also has pretty extensive experiments, and an unusually detailed discussion of the policy/legal side of things.

**Audience:**

Yes

**Audience Explanation:**

Members of the TMLR community are certainly interested in large language models, their pre-training datasets, and the curation thereof. I suspect there also is a small but significant community interested in the legal aspects that this submission addresses.

**Claims And Evidence:**

Yes

**Claims Explanation:**

All technical claims regarding the quality and effectiveness of this pre-training dataset were verified on standard benchmarks. One reviewer asked for more "modern"/more complex benchmarks and the concerns were addressed in the rebuttal. Non-technical claims regarding licensing status are discussed in significant detail in the paper, including guidance for practitioners.

---

> ### Author Response · Authors · 2026-04-21
>
> Dear Action Editor, dear reviewers,
>
> following the positive Action Editor decision, we submitted the camery ready version, incorporating outcomes from the rebuttal exchange. Thanks a lot for the work and time spent for our submission, looking forward any further exchange.
>
> Best,
>
> Authors